# Neural Graduated Assignment for Maximum Common Edge Subgraphs

**Chaolong Ying**[†]**, Yingqi Ruan**[†]**, Xuemin Chen**[‡]**, Yaomin Wang**[†]**, Tianshu Yu**[†*]

[†] School of Data Science, The Chinese University of Hong Kong, Shenzhen
[‡] School of Science and Engineering, The Chinese University of Hong Kong, Shenzhen
{chaolongying,yingqiruan,xueminchen,yaominwang}@link.cuhk.edu.cn,
yutianshu@cuhk.edu.cn

## Abstract

The Maximum Common Edge Subgraph (MCES) problem is a crucial challenge with significant implications in domains such as biology and chemistry. Traditional approaches, which include transformations into max-clique and search-based algorithms, suffer from scalability issues when dealing with larger instances. This paper introduces "Neural Graduated Assignment" (NGA), a simple, scalable, unsupervised-training-based method that addresses these limitations. Central to NGA is stacking of differentiable assignment optimization with neural components, enabling high-dimensional parameterization of the matching process through a learnable temperature mechanism. We further theoretically analyze the learning dynamics of NGA, showing its design leads to fast convergence, better exploration-exploitation tradeoff, and ability to escape local optima. Extensive experiments across MCES computation, graph similarity estimation, and graph retrieval tasks reveal that NGA not only significantly improves computation time and scalability on large instances but also enhances performance compared to existing methodologies. The introduction of NGA marks a significant advancement in the computation of MCES and offers insights into other assignment problems.

## 1 Introduction

**Background.** The Maximum Common Edge Subgraph (MCES) problem is a cornerstone task in combinatorial optimization (Ndiaye & Solnon, 2011), particularly significant within the realms of biology and chemistry (Ehrlich & Rarey, 2011). As a variant of the broader Maximal Common Subgraph (MCS) problem (Bunke & Shearer, 1998), MCES stands alongside the Maximal Common Induced Subgraph (MCIS) in its complexity and utility. Efficiently solving the MCES problem at scale is of both theoretical and practical significance in this context. For instance, in drug discovery, identifying MCES between molecules can reveal shared pharmacological properties. In cybersecurity, comparing network traffic graphs via MCES enables the detection of recurring attack patterns (Ehrlich & Rarey, 2011). Established approaches for solving MCES have been successfully integrated into widely-used cheminformatics libraries such as RDKit (Bento et al., 2020) and Molassembler (Sobez & Reiher, 2020), which have become indispensable in both industrial and academic research.

**Problem & Existing Challenges.** MCES involves identifying a subgraph that contains the maximum number of edges common to both input graphs. An example is in Figure 1. It is inherently NP-complete (Garey & Johnson, 1979), posing significant computational challenges in terms of scalability and efficiency. Historically, it has been tackled through transformations into the maximum clique problem (Bomze et al., 1999) or search-based algorithms (Raymond et al., 2002; McCreesh et al., 2017; 2016). However, these traditional methods

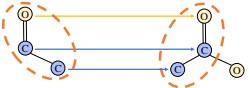

Figure 1: For two labeled graphs, e.g. molecular graphs, the MCES is highlighted in circle and the node correspondences are encoded in arrowed lines.

---

[*]Corresponding author

often struggle with large-scale graphs: transformations can introduce additional computational overhead, while search-based branch-and-bound approaches suffer from exponential scaling, rendering them impractical for complex instances. This challenging landscape underscores the need for novel techniques that can deliver efficient and reliable solutions to the MCES problem without compromising accuracy.

**Our approach.** To address the challenges of MCES, we propose **Neural Graduated Assignment** (NGA) – a novel neuralized optimization framework designed to approximate MCES solutions in polynomial time. Inspired by annealing mechanism in statistical physics (Gold & Rangarajan, 1996), our approach seeks to find an assignment matrix that identifies node correspondences between the two graphs, such that the number of preserved edges – i.e., edges that exist between matched node pairs in both graphs – is maximized. NGA tackles the aforementioned intrinsic challenges in the MCES problem by: (**1**) constructing an **Association Common Graph** (ACG) and formulating MCES as a Quadratic Assignment Problem (QAP), which ensures the extraction of exact common subgraphs between input graphs; (**2**) iteratively updating the learned assignments through a high-dimensional and learnable temperature parameterization; and (**3**) optimizing this formulation via unsupervised training[1], thereby eliminating the need for training data. During optimization, the learned assignments correspond to a current identified common edge subgraph that serves as a *lower bound*, which is progressively and monotontically improved toward the optimal MCES. An illustrative example of such improved lower bound is provided in Figure 9 of the Appendix.

**Theory.** Despite the simplicity, we theoretically justify the superiority of NGA by showing: (**1**) how NGA updates around local optima; (**2**) the implicit exploration-exploitation mechanism impacts the optimization trajectory; and (**3**) the accelerating effect on the convergence. By embedding these operations within a neural context, NGA achieves strong scalability, adaptability, and seamless integration with machine learning frameworks. This theoretical underpinning lays the groundwork for a new class of algorithms capable of rethinking assignment problems from a neural perspective.

**Results & Contribution.** To validate the efficacy of our approach, we conduct a series of challenging experiments across various settings, including large-scale MCES computation, graph similarity estimation, and structure-based graph retrieval. The results consistently demonstrate the superior efficiency and performance of our method compared to existing ones. These findings not only highlight the potential of our approach in advancing MCES computation but also suggest its applicability to diverse domains. We summarize our contributions as follows:

- **A novel and strong approach.** We for the first time formulate the MCES problem via the construction of an ACG, and based on this formulation, we propose NGA – the first neural-style algorithm to approximate the MCES solution efficiently in polynomial time without relying on exhaustive exploration of the solution space, delivering superior performance.

- **Training-data-free.** NGA operates in a fully unsupervised manner, eliminating the need for supervision signals. In cases where enumeration or search of the exact solution becomes computationally infeasible, our approach provides efficient approximations.

- **Theoretical analysis.** We provide the first theoretical analysis on the behavior of dynamic and parameterized temperature in NGA, shedding light on its behavior at local optima, convergence properties and adaptive dynamics.

- **Interpretability, scalability, and versatility.** NGA is inherently interpretable, producing explicit MCES structures. It also exhibits strong scalability, making it suitable for large-scale graph data. Furthermore, NGA can be extended to tasks such as graph similarity computation and graph retrieval, highlighting its broad applicability.

## 2   RELATED WORK

**Efforts on Solving MCS.**   MCS has been extensively studied in recent research, either by reduction to the maximum clique problem (McCreesh et al., 2017) or by directly identifying matching pairs within the original problem formulation. McSplit (McCreesh et al., 2017), which belongs to the latter category, introduces an efficient branch-and-bound algorithm leveraging node labeling and

---

[1]Unsupervised training means that no information about the MCES itself—such as what the MCES is or the MCES size—is required during training, including in the loss function (Schuetz et al., 2022).

partitioning techniques to reduce memory and computational requirements during the search. Building on this, GLSearch (Bai et al., 2021) employs a GNN-based Deep Q-Network to select matched node pairs instead of relying on heuristics, significantly improving search efficiency. For the MCES problem, RASCAL (Raymond et al., 2002) proposes a branching-based search method to solve its maximum clique formulation, incorporating several heuristics to accelerate the search process. Additionally, MCES has been formulated as an integer programming problem and solved using a branch-and-cut enumeration algorithm (Bahiense et al., 2012). However, these methods suffer from high computational costs, making them less scalable for solving challenging MCS problems.

**Efforts on Graph Similarity Computation and Graph Retrieval.** Recent efforts have explored the application of MCS in various tasks, particularly in graph similarity computation and graph retrieval. For assessing graph similarity, one established approach is the use of Graph Edit Distance (GED) (Zhao et al., 2013; Zheng et al., 2013). Alternatively, MCS provides a robust method for evaluating the structural similarity of graphs, especially in molecular applications. For example, SimGNN (Bai et al., 2019) integrates node-level and graph-level embeddings to compute a similarity score, while GMN (Li et al., 2019) introduces a novel cross-graph attention mechanism for learning graph similarity. Similarly, INFMCS (Lan et al., 2024) presents an interpretable framework that implicitly infers MCS to learn graph similarity. In the domain of graph retrieval, where the goal is to locate the most relevant or similar graphs in a database based on a query graph, various model architectures have been proposed. For instance, ISONET (Roy et al., 2022b) employs subgraph matching within an interpretable framework to calculate similarity scores. Furthermore, XMCS (Roy et al., 2022a) proposes late and early interaction networks to infer MCS as a similarity metric, offering competitive performance in terms of both accuracy and computational speed.

## 3 PRELIMINARY

In this section, we briefly review the background of this topic, as well as elaborate on the notations. Additional background can be found in Appendix A.

Let $G = (\mathcal{V}, \mathcal{E}, \mathcal{A}, \mathbf{H}, \mathbf{E})$ be an undirected and labeled graph with $n$ nodes, where $\mathcal{V}$ is the node set, $\mathcal{E}$ is the edge set, $\mathcal{A} \in \{0,1\}^{n \times n}$ is the adjacency matrix, $\mathbf{H} \in \mathbb{R}^{n \times \cdot}$ is the node feature matrix, and $\mathbf{E} \in \mathbb{R}^{n \times n \times \cdot}$ is the edge feature matrix. In a labeled graph, nodes and edges features refer to their labels. For example, in molecular graphs, atom types and bond types are considered as labels.

**Maximum Common Edge Subgraph.** Two graphs, $G_1$ and $G_2$, are isomorphic if there exists a bijective mapping between their nodes such that any two nodes in $G_1$ are connected by an edge if and only if their corresponding images in $G_2$ are also connected. A common subgraph of two graphs $G_1$ and $G_2$ is a graph $G_{12}$ that is isomorphic to a subgraph of $G_1$ and a subgraph of $G_2$. Although there are possibly many common subgraphs between two graphs, our focus will be on the MCES (Bahiense et al., 2012), which is a subgraph with the maximal number of edges common to both $G_1$ and $G_2$.

**Quadratic Assignment Problem.** Given two graphs $G_1$ and $G_2$, the goal is to find a hard assignment matrix $\mathbf{P} \in \{0,1\}^{n_1 \times n_2}$ that maximizes a compatibility function while adhering to row and column constraints. The optimization problem is formalized as:

$$
\begin{aligned}
\max_{\mathbf{P}} \quad & \text{vec}(\mathbf{P})^{\top} \mathbf{A} \text{vec}(\mathbf{P}) \\
\text{s.t.} \quad & \mathbf{P} \in \{0,1\}^{n_1 \times n_2}, \mathbf{P} \mathbf{1}_{n_2} = \mathbf{1}_{n_1}, \mathbf{P}^{\top} \mathbf{1}_{n_1} \leq \mathbf{1}_{n_2},
\end{aligned}
\tag{1}
$$

where $\mathbf{A}$ is the affinity matrix derived from the structural information of $G_1$ and $G_2$ and $\mathbf{1}_n$ is a column vector of length $n$ whose elements are all equal to 1. A common practice is to relax $\mathbf{P}$ into a soft assignment matrix $\mathbf{S} \in [0,1]^{n_1 \times n_2}$, allowing continuous values, and optimizes the following relaxed objective:

$$
\max_{\mathbf{S}} \quad \text{vec}(\mathbf{S})^{\top} \mathbf{A} \text{vec}(\mathbf{S})
\tag{2}
$$

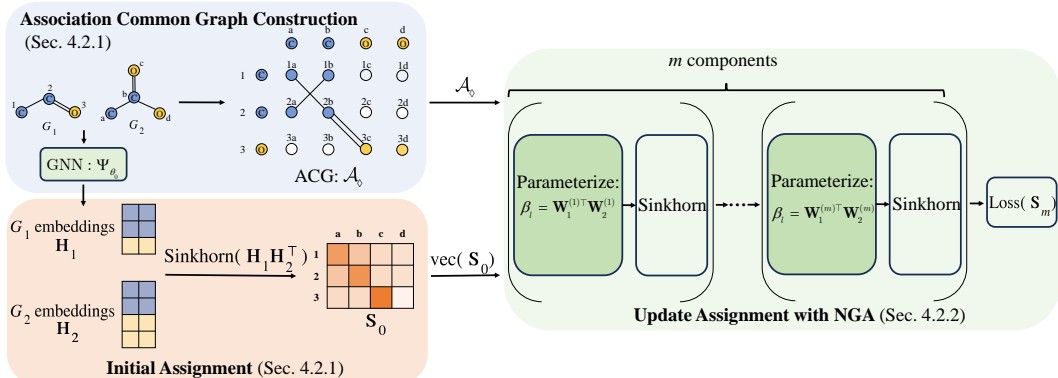

Figure 2: An overview of Neural Graduated Assignment.

# 4 METHODOLOGY

## 4.1 OVERVIEW

The MCES problem is intricately connected to graph matching, as both involve finding correspondences between the structures of two graphs. Specifically, the MCES problem aims to find the largest subgraph that is isomorphic to subgraphs in both input graphs. This can be viewed as a special case of *partial graph matching*, where the matched pairs of nodes and edges constitute common subgraphs. Typically, graph matching seeks to find an assignment matrix $\mathbf{P}$ as follows (Grohe et al., 2018; Liu et al., 2023; Ying et al., 2025):

$$\operatorname{argmin}_{\mathbf{P}} \left\| \mathcal{A}_1 - \mathbf{P}\mathcal{A}_2\mathbf{P}^\top \right\|_F^2. \tag{3}$$

However, a prominent challenge for the MCES of labeled graphs is ensuring that the corresponding nodes and edges in the common subgraphs have compatible labels, which are not considered in typical graph matching problems. One possible solution would be to add a penalty term for incompatible labels. However, directly optimizing Eq. (3) with penalty term is intractable, as it cannot guarantee that the obtained subgraph is a valid common subgraph. To this end, we present a novel approach centered around the construction of an ACG, which enables the extraction of exact common subgraphs. Building on this framework, we propose a new formulation for the MCES problem. This formulation leverages the ACG and facilitates optimization by learning an assignment between the nodes of the two input graphs, resulting in an efficient and scalable solution for MCES computation. An overview of our method is shown in Fig. 2. In the following, Section 4.2.1 presents the construction of the ACG and formulates the MCES problem as a QAP, Section 4.2.2 describes the proposed NGA method, and Section 4.2.3 introduces the Gumbel sampling strategy used during inference.

## 4.2 UNSUPERVISED TRAINING OF MCES

### 4.2.1 ACG FOR LEARNING THE CORRESPONDENCES

To efficiently explore solutions of MCES, we proposed to construct an ACG, where compatible nodes and edges can be identified. We define the ACG of $G_1$ and $G_2$ as $G_1 \Diamond G_2$. It is constructed on the node set $\mathcal{V}(G_1 \Diamond G_2) = \mathcal{V}(G_1) \times \mathcal{V}(G_2)$ where the respective node labels are compatible and two nodes $(u_i, v_i)$ and $(u_j, v_j)$ being adjacent whenever the three conditions are met together

$$(u_i, u_j) \in \mathcal{E}(G_1), \ (v_i, v_j) \in \mathcal{E}(G_2), \ \omega(u_i, u_j) = \omega(v_i, v_j), \tag{4}$$

where $\omega(u_i, u_j) = \omega(v_i, v_j)$ indicates that the labels of nodes and edges are compatible. The design of ACG leads to a well-behaved property as follow.

**Proposition 1.** *Denote the node set of $G_1 \Diamond G_2$ as $\mathcal{V}_\Diamond = \{(u, v) | u \in \mathcal{V}(G_1), v \in \mathcal{V}(G_2)\}$. Consider a set of subgraphs of $G_1 \Diamond G_2$ where each node of $G_1$ and $G_2$ is selected at most once, i.e. any two nodes $(u_i, v_i)$ and $(u_j, v_j)$ in this set satisfy $u_i \neq u_j$ and $v_i \neq v_j$. **Any** subgraph in this set is a valid common subgraph of $G_1$ and $G_2$, and finding the MCES of two graphs is reduced to finding the largest subgraph in this set.*

Since only the common edges are included in the ACG, Proposition 1 shows that we can always extract common subgraphs from the constructed ACG. Besides, the use of ACG allows any assignment to be mapped to a common subgraph, thereby enhancing interpretability. Formal proof of Proposition 1 can be found in Appendix B. Considering the adjacency matrix $\mathcal{A}_\diamond$ of $G_1 \diamond G_2$ as the affinity matrix, we can then give the QAP formulation of objective $J$ for MCES as:

$$J(\mathbf{S}) = \text{vec}(\mathbf{S})^\top \mathcal{A}_\diamond \text{vec}(\mathbf{S}). \tag{5}$$

We model the initial node-to-node correspondence in close analogy to related approaches (Bai et al., 2019; Fey et al., 2019) by computing the node similarities of input graphs $G_1$ and $G_2$. Specifically, given latent node embeddings $\mathbf{H}_1 = \mathbf{\Psi}_{\theta_0}(\mathcal{A}_1, \mathbf{X}_1, \mathbf{E}_1)$ and $\mathbf{H}_2 = \mathbf{\Psi}_{\theta_0}(\mathcal{A}_2, \mathbf{X}_2, \mathbf{E}_2)$ computed by a shared neural network $\mathbf{\Psi}_{\theta_0}$, the initial soft correspondences are obtained as

$$\hat{\mathbf{S}}^{(0)} = \mathbf{H}_1 \mathbf{H}_2^\top \in \mathbb{R}^{n_1 \times n_2}, \qquad \mathbf{S}^{(0)} = \text{Sinkhorn}(\hat{\mathbf{S}}_0) \in [0,1]^{n_1 \times n_2}, \tag{6}$$

where $\text{Sinkhorn}(\cdot)$ is a normalization operator to obtain double-stochastic assignment matrices (Sinkhorn & Knopp, 1967). In our implementation, $\mathbf{\Psi}_{\theta_0}$ is a GNN to obtain permutation equivariant node representations (Hamilton et al., 2017).

### 4.2.2 NEURAL GRADUATED ASSIGNMENT

Since many nodes have the same labels in molecular graphs, the initial assignment is a soft probability distribution where many entries have moderate probabilities, which leaves significant ambiguity in the assignments. Traditional assignment-based optimization methods often rely on a predefined temperature parameter $\beta$ to iteratively refine solutions by controlling the scale of probabilities or soft assignments. However, using a fixed or manually scheduled temperature introduces two key limitations: **(1) Manual Tuning Overhead**: Selecting an optimal schedule or fixed value for $\beta$ often requires exhaustive tuning, depending on the specific task or dataset. **(2) Limited Adaptability:** A fixed or scheduled $\beta$ is static and cannot dynamically adapt to the complexity of individual problems or local structures within data.

To address these limitations, we propose NGA, an end-to-end trainable framework in which we use learnable parameters as the temperature schedule. The NGA architecture is summarized in Alg. 1, and the overall training and inference framework are summarized in Alg. 2. NGA maintains the iterative refinement process. The key innovation lies in replacing the scalar temperature parameter with a parameterized form: $\beta_l = \mathbf{W}_1^{(l)\top} \mathbf{W}_2^{(l)}$ at the $l$th iteration layer, where $\mathbf{W}_1^{(l)}, \mathbf{W}_2^{(l)} \in \mathbb{R}^{d \times 1}$ are learnable weights. By the model to optimize temperature directly as part of the learning process, our method adapts dynamically to the structure of the problem and the stage of optimization, eliminating the dependence on manual tuning while achieving superior performance.

---

**Algorithm 1** NGA Architecture

1: **Input:** The adjacency matrix of ACG $\mathcal{A}_\diamond$, inital assignment matrix $\mathbf{S}^{(0)}$, number of iterations $m$, learnable parameters $\mathbf{W}_1^{(l)}, \mathbf{W}_2^{(l)}$ for $l \in \{1, ..., m\}$
2: **Output:** refined assignment matrix $\mathbf{S}$
3: **for** $l = 1$ **to** $m$ **do**
4:     $\text{vec}(\mathbf{S}^{(l)}) \leftarrow \mathcal{A}_\diamond \text{vec}(\mathbf{S}^{(l-1)})$
5:     $\mathbf{S}^{(l)} \leftarrow \exp((\mathbf{W}_1^{(l)\top} \mathbf{W}_2^{(l)}) \mathbf{S}^{(l)})$
6:     $\mathbf{S}^{(l)} = \text{Sinkhorn}(\mathbf{S}^{(l)})$
7: **end for**
8: Output $\mathbf{S} = \mathbf{S}^{(m)}$

---

**Algorithm 2** Model Training and Inference

1: **Input:** Molecular graphs $G_1$ and $G_2$
2: **Output:** MCES of $G_1$ and $G_2$
3: Build ACG $G_1 \diamond G_2$
4: // Training Stage
5: **for** epoch $= 1, 2, 3, ...$ **do**
6:     Init assignment $\mathbf{S}^{(0)}$ with Eq. (6)
7:     Refine $\mathbf{S}^{(0)}$ to $\mathbf{S}$ with Alg. 1
8:     Backpropagation w.r.t. loss in Eq. (5)
9: **end for**
10: // Inference Stage
11: Decode $\mathbf{P}$ with $\mathbf{P} = \text{Hungarian}(\mathbf{S})$
12: Get MCES w.r.t. $\mathbf{P}$ via Eq. (7)

---

### 4.2.3 GUMBEL SAMPLING FOR OPTIMIZATION

Since $\mathbf{S}$ is a relaxed from $\mathbf{P}$, the Hungarian algorithm (Burkard et al., 2012) is commonly employed as a deterministic post-processing step to bridge the gap between them, i.e. $\mathbf{P} = \text{Hungarian}(\mathbf{S})$. The adjacency matrix of predicted common subgraph $\mathcal{A}_{\text{pred}}$ is obtained by:

$$\mathcal{A}_{\text{pred}} = \left(\text{vec}(\mathbf{P})\text{vec}(\mathbf{P})^\top\right) \odot \mathcal{A}_\diamond, \tag{7}$$

where $\odot$ is the Hadamard product of matrices.

From a probabilistic perspective, $\mathbf{S}$ represents a latent distribution of assignment matrices. Assignment with the highest probability is selected by Hungarian algorithm. However, there may be better solutions within the distribution, especially when the quality of solutions can be easily assessed. Therefore, we switch to Gumbel-Sinkhorn (Mena et al., 2018) by substituting Eq. (6) with

$$\mathbf{S}^{(0)} = \text{Sinkhorn}(\hat{\mathbf{S}}^{(0)} + g), \tag{8}$$

where $g$ is sampled from a standard Gumbel distribution with the cumulative distribution function. The Gumbel term models the distribution of extreme values derived from another distribution. With Eq. (8), we can sample repeatedly *a batch of assignment matrices* from the original distribution in a differentiable way. This property actively benefits solution space exploration and makes it easier to find the optimal solution. These sampled assignment matrices are discretized by Hungarian algorithm and evaluated by Eq. (7). The best-performing solution among them is chosen as the final solution. The balance between exploration and speed can be adjusted by tuning the number of Gumbel samples.

## 5 THEORETICAL ANALYSIS

In this section, we investigate the behavior of NGA from theoretical aspects, particularly answering subsequent essential questions: **1**) How NGA escapes from local optima in Theorem 1; **2**) Why NGA demonstrates better convergence and performance over standard GA in Proposition 2. Upon these theoretical findings, we argue that NGA offers an adaptive exploration-exploitation mechanism, leading to well-behaved learning dynamics.

**Theorem 1.** *Given a local optimum* $\mathbf{S}^{(l)}$*, the change* $\Delta J = J(\mathbf{S}^{(l+1)}) - J(\mathbf{S}^{(l)})$ *satisfies*

$$\Delta J = \frac{1}{2}(\sum_i \lambda_i - \sum_j \mu_j) + \beta_l \cdot \text{Var}(h) + O(|\beta_l|^2), \tag{9}$$

*where* $h_{ij} = [\mathcal{A}_\Diamond \text{vec}(\mathbf{S}^{(l)})]_{ij}$*,* $\text{Var}(h)$ *is the variance,* $J(\mathbf{S}^{(l)}) = \text{vec}(\mathbf{S}^{(l)})^\top \mathcal{A}_\Diamond \text{vec}(\mathbf{S}^{(l)})$ *is the objective,* $\lambda_i$ *and* $\mu_j$ *are the Lagrange multipliers.*

**Proposition 2.** *Under typical gradient descent conditions where the gradient pushes* $\beta_l$ *in a consistent direction for multiple iterations, the product parameterization* $\beta_l = \mathbf{W}_1^\top \mathbf{W}_2$ *can adaptively adjust the learning rate and induces an accelerating effect on the magnitude of updates to* $\beta_l$*.*

The proof of Theorem 1 can be found in Appendix E. This theorem quantifies the objective change around a local optima w.r.t. $\beta_l$. In general, $\Delta J$ has an additional constant term related to Lagrange multipliers, but still decreases through the variance term when $\beta_l < 0$. This demonstrates that NGA's update is capable of escaping local optimum under mild conditions.

Another contributing factor to the effectiveness of NGA lies in the gradient update aspect. As shown in Proposition 2, the product parameterization $\beta_l = \mathbf{W}_1^{(l)\top} \mathbf{W}_2^{(l)}$ can adaptively adjust the learning rate and leads to faster convergence than scalar parameterization. This makes the training process more stable and efficient. This adaptive behavior is particularly beneficial when addressing complex, high-dimensional data. Detailed theoretical and empirical validations can be found in Appendix D.2.

We then analyze how switching $\beta_l$ positive and negative interchangeably can help the optimization. To this end, we make the following definition:

**Definition 1.** In Alg. 1, let $\beta_l = \mathbf{W}_1^{(l)\top} \mathbf{W}_2^{(l)}$, then we define two phases according to the value of $\beta_l$: 1. Exploration phase: $\mathcal{E} = \{l | \beta_l < 0\}$; 2. Exploitation phase: $\mathcal{S} = \{l | \beta_l > 0\}$.

Appendix D.1 provides a proof explaining why the sign of $\beta_l$ represents these two processes, respectively. Imposing a negative and learnable $\beta_l$ in NGA grants the model the ability to balance exploration and exploitation. By allowing the two phases, NGA effectively incorporates exploration and exploitation into the optimization, avoiding premature convergence and enabling the identification of better solutions.

## 6 EXPERIMENTS

We evaluate the performance of our proposed method on MCES problems by comparing it with several baselines across three widely used molecular datasets, covering diverse graph structures. Our results show that NGA significantly outperforms search-based methods in computational efficiency, achieving comparable or better accuracy while reducing runtime by several orders of magnitude. This allows our method to scale to larger graphs that are infeasible for traditional search, highlighting its effectiveness and scalability in real-world applications.

Beyond solving the MCES problem, we further examine its practical impact on two related tasks: graph similarity computation and graph retrieval. By effectively solving MCES, our method achieves strong performance in these tasks, offering a reliable basis for similarity measurement and retrieval. This confirms that NGA not only addresses MCES effectively but also benefits related applications.

### 6.1 EXPERIMENTAL SETUP

**Baselines.** The MCES problem is NP-complete (Garey & Johnson, 1979; Raymond et al., 2002), meaning no known polynomial-time solutions exist. While search-based algorithms can find exact solutions with enough computational resources, they quickly become infeasible for large graphs. Non-search-based methods aim to approximate solutions efficiently without exhaustive search. We compare our method with search-based RASCAL (Raymond et al., 2002) and Mcsplit (McCreesh et al., 2017), and use Graduated Assignment (GA) (Rangarajan et al., 1996b; Gold & Rangarajan, 1996) and Gurobi solver as baselines since we can formulate MCES as a QAP. Similar to these solvers, our NGA tries to solve each MCES instance case by case. We also adapt supervised learning method NGM (Wang et al., 2021) and unsupervised learning method GANN-GM (Wang et al., 2023) for comparison. For the graph similarity and retrieval tasks, we compare with state-of-the-art baselines: NeuroMatch (Lou et al., 2020), SimGNN (Bai et al., 2019), GMN (Li et al., 2019), XMCS (Roy et al., 2022a), and INFMCS (Lan et al., 2024). Appendix C.2 details baseline choices and configurations.

For fair comparisons, supervised learning approaches are trained for a total of 200 epochs. This duration is selected to provide sufficient learning time for the models to converge, ensuring robust evaluation of their performance. For unsupervised learning and search methods, we follow previous works (Bai et al., 2021) to set a practical time budget of 60 seconds for each instance. This constraint reflects real-world scenarios where computation time is limited and must be optimized for efficiency. While search methods are capable of identifying exact solutions given unlimited computational resources, this assumption is impractical for handling large graph pairs in real-world applications.

**Datasets.** To evaluate our model across diverse domains, we use several graph datasets commonly employed in graph-related tasks: AIDS and MCF-7 from TU Dataset (Morris et al., 2020), and MOLHIV from OGB (Hu et al., 2020), covering various molecular structures. Ground truth is obtained using exact solvers (Raymond et al., 2002), with solution time and dataset statistics in Appendix C.1. While exact MCES solutions are feasible for small graphs, they become intractable for large ones. Unlike prior works that limit graphs to 15 nodes (Bai et al., 2019; Lan et al., 2024), we focus on harder instances by selecting molecules with over 30 atoms, and randomly sample 1000 graph pairs per dataset for MCES and similarity evaluation. For graph retrieval, we use 100 query and 100 target graphs, yielding 10,000 query-target pairs. This setup enables a realistic assessment of scalability and efficiency.

**Evaluation.** For MCES and graph similarity tasks, we split the dataset into 80% training, 10% validation, and 10% testing. For graph retrieval, this split is applied to the query graphs. Supervised methods are trained on the training set, with hyperparameters tuned on the validation set and evaluated on the test set. Other methods are directly evaluated on the test set. Since all methods yield common subgraphs, MCES accuracy is measured by the percentage of the common graph size, defined as:

$$Acc = \frac{|\mathcal{E}\left(\mathrm{CS}(G_1, G_2)\right)|}{|\mathcal{E}\left(\mathrm{MCES}(G_1, G_2)\right)|}. \tag{10}$$

As for the graph similarity computation, the Johnson similarity (Johnson, 1985) is calculated as

$$\mathrm{sim}(G_1, G_2) = \frac{\left(|\mathcal{V}\left(G_{12}\right)| + |\mathcal{E}\left(G_{12}\right)|\right)^2}{\left(|\mathcal{V}\left(G_1\right)| + |\mathcal{E}\left(G_1\right)|\right) \cdot \left(|\mathcal{V}\left(G_2\right)| + |\mathcal{E}\left(G_2\right)|\right)}, \tag{11}$$

where $G_{12}$ is the MCES between graphs $G_1$ and $G_2$, and we use Root Mean Square Error (RMSE) as the evaluation metric. For graph retrieval, we evaluated all models using three metrics: Mean Reciprocal Rank (MRR), Precision at 10 (P@10), and Mean Average Precision (MAP). These metrics collectively provide insights into the retrieval capability from various perspectives.

## 6.2 Implementation Details

**Hyperparameters.** In our method, we use two neural networks: the first $\mathbf{\Psi}_{\theta_0}$ is an 8-layer Graph Convolutional Network (GCN) (Kipf & Welling, 2016), which utilizes 32 feature channels. In our NGA model, the number of iterations $m$ is set to 4 and the hidden dimension of learnable weights is set to 32. For the Sinkhorn layer, we set the number of iterations to 20 to allow for sufficient optimization. The GumbelSinkhorn layer is configured with 10 times sampling. The model is trained using the Adam (Kingma, 2014) optimizer with a learning rate of 0.001. These hyperparameter settings were chosen to balance model performance and computational efficiency, allowing for effective training and convergence across the evaluated tasks.

**Training Details of NGA.** The training and inference processes are summarized in Alg. 2. Similar to conventional MCES solvers (McCreesh et al., 2017; Raymond et al., 2002), our method addresses one pair of molecular graphs from the test set at a time. During the training stage, our method learns to optimize the objective in Eq. (5). In the inference stage, the assignment matrix $\mathbf{S}$ learned at the training stage is transformed into an assignment matrix $\mathbf{P}$, which is then used to compute the MCES result with Eq. (7). Specifically, when Gumbel sampling is employed for optimization, Eq. (6) in line 6 is replaced with Eq. (8). Under this setup, in lines 11-12, $M$ assignment matrices are generated using $M$ sampled Gumbel noises, leading to $M$ predicted MCES results. These results are evaluated using Eq. (5), and the result with the best objective value is retained as the final output.

**Model Configuration.** Our implementation of NGA and the baseline methods was conducted using Pytorch and PyG (Paszke et al., 2017; Fey & Lenssen, 2019). Most baseline methods were adapted from their official source code repositories to ensure consistency with their original implementations. However, for INFMCS, no official source code was publicly available, so we implemented this method from scratch, closely following the algorithmic details and hyperparameter settings described in the original paper (Lan et al., 2024). Our implementation was rigorously validated against the reported results in the literature to ensure correctness. Given the molecular nature of our dataset, we utilized the atom encoder and bond encoder from the Open Graph Benchmark library[2] for feature encoding in all models employing GNN as their backbone. This ensured a fair and consistent representation of molecular data across all GNN-based baselines and our proposed method. The experiments are conducted using an AMD EPYC 7542 CPU and a single NVIDIA 3090 GPU.

## 6.3 Solving MCES Problems

The key property of NGA is its ability to approximate solutions in polynominal time complexity. As shown in Table 1, our model outperforms baselines on all datasets. Our results are very close to optimal in all datasets, which shows that NGA can effectively approximate solutions within limited time. We have additionally analyzed how often the MCES found by NGA matches the optimal solution in Appendix G.1. The supervised graph matching method does not perform well, possibly because the MCES problem is inherently multi-modal, meaning it has multiple equally optimal solutions. It is thus difficult for supervised learning method to learn some patterns from multiple ground truth node matching signals. The search method, on the other hand, suffers from scalability issues due to the NP-complete nature of the MCES problem. In the following part, we provide several case studies that investigate the effects of increasing the time budget on performance.

We selected several cases for detailed analysis. The largest MCES solution found thus far as time grows is illustrated in Figure 8 (in Appendix G.2). The visualizations of the MCES structures identified by our NGA method and the second-best method, RASCAL, are shown in Figure 3 and Figure 10 (in Appendix G.2). From these figures, it is evident that our method can find better solutions in less time, demonstrating its superiority over other competitive baselines.

---

[2]`https://github.com/snap-stanford/ogb`

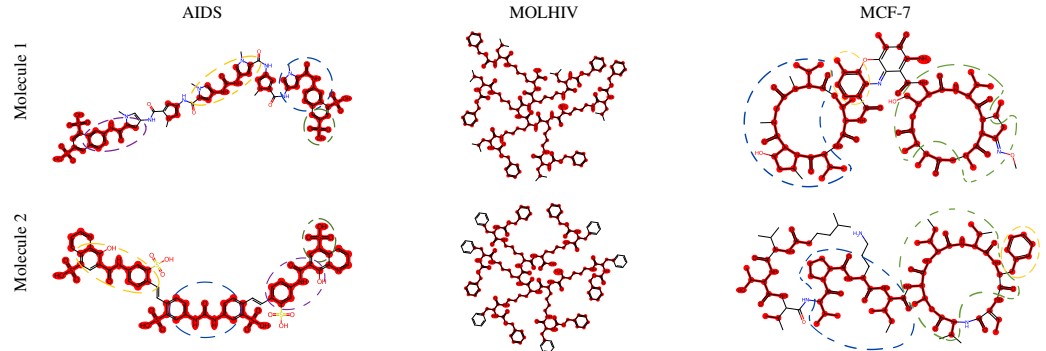

Figure 3: Visualization of MCES with NGA. Matched atoms and bonds are highlighted in red, with circles of the same color indicating correspondences.

Table 3: Comparison of graph retrieval results in terms of three evaluation metrics.

| | Mean Reciprocal Rank (MRR) ↑ | | | Precision at 10 (P@10) ↑ | | | Mean Average Precision (MAP) ↑ | | |
|---|---|---|---|---|---|---|---|---|---|
| | AIDS | MOLHIV | MCF-7 | AIDS | MOLHIV | MCF-7 | AIDS | MOLHIV | MCF-7 |
| NeuroMatch | 0.676 | 0.540 | 0.393 | 0.870 | 0.870 | 0.730 | 0.911 | 0.779 | 0.739 |
| SimGNN | 0.419 | 0.207 | 0.468 | 0.870 | 0.830 | 0.850 | 0.875 | 0.726 | 0.703 |
| GMN | 0.238 | 0.291 | 0.279 | 0.400 | 0.310 | 0.540 | 0.683 | 0.699 | 0.652 |
| XMCS | 0.575 | 0.192 | 0.530 | 0.850 | 0.770 | 0.730 | 0.890 | 0.712 | 0.753 |
| INFMCS | 0.560 | 0.261 | 0.324 | 0.800 | 0.540 | 0.840 | 0.914 | 0.489 | 0.788 |
| NGA (ours) | **0.844** | **0.806** | **0.803** | **0.890** | **0.980** | **0.890** | **0.974** | **0.951** | **0.966** |

Table 1: Results of Accuracy $(\%)$ ↑ for MCES.

| Dataset | AIDS | MOLHIV | MCF-7 |
|---|---|---|---|
| RASCAL | 90.67 | 88.74 | 90.26 |
| FMCS | 60.63 | 67.99 | 71.54 |
| Mcsplit | 61.32 | 72.74 | 69.23 |
| GLSearch | 43.47 | 41.12 | 42.08 |
| NGM | 33.34 | 52.21 | 42.38 |
| GANN-GM | 49.76 | 72.18 | 63.47 |
| GA | 70.99 | 71.09 | 72.92 |
| Gurobi | 74.52 | 78.67 | 80.76 |
| NGA (ours) | **98.64** | **99.20** | **97.94** |

Table 2: Results of MSE $(\times 10^{-3})$ ↓ for graph similarity.

| Dataset | AIDS | MOLHIV | MCF-7 |
|---|---|---|---|
| NeuroMatch | 13.92 | 19.17 | 17.34 |
| SimGNN | 15.42 | 14.38 | 20.29 |
| GMN | 39.38 | 29.77 | 24.03 |
| XMCS | 36.42 | 21.47 | 32.22 |
| INFMCS | 25.18 | 16.39 | 31.12 |
| NGA (ours) | **1.13** | **0.66** | **0.99** |

## 6.4 GRAPH SIMILARITY COMPUTATION AND GRAPH RETRIEVAL

In the graph similarity computation experiments, we compare NGA with several baselines. As shown in Table 2, our method surpasses other compared methods by an order of magnitude. Although some methods, such as XMCS and INFMCS, learn similarity by implicitly inferring the size of MCES, they disregard the detailed structure of the MCES solution. Our method can learn to generate precise approximation of the MCES structure, leading to lower similarity computation errors.

The graph retrieval results are summarized in Table 3, where three evaluation metrics are employed to assess performance from different perspectives. As another application of MCES, graph retrieval emphasizes the relative similarity between graphs rather than the precise computation of graph similarity. However, in cases where a group of target graphs share similar characteristics with the query graph, distinguishing their differences may require analyzing the details of common substructures. The experimental results demonstrate that our method is stable in retrieval scenarios.

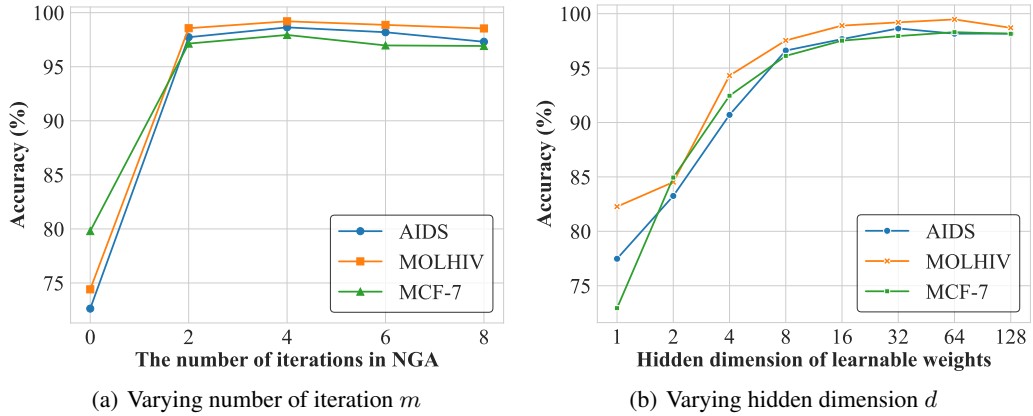

(a) Varying number of iteration $m$

(b) Varying hidden dimension $d$

Figure 4: How NGA performs on MCES with different parameters.

### 6.5 MORE ANALYSIS

In this section, we first analyze the hyperparameters to better understand their impact on the performance of NGA. Details of the ablation study are provided in Appendix F, and the analysis of the number of Gumbel samples is shown in Appendix G.4. Here, we summarize the impact of the number of iterations $m$ and the hidden dimensions $d$ in Figure 4.

When $m = 0$, our method excludes the NGA process entirely, relying solely on similarities computed by the graph feature encoder (Eq. (6)). In this scenario, we observe a sharp decrease in performance, highlighting the critical role of NGA in achieving strong results. Unlike traditional GA, which requires dozens of iterations to converge, our method delivers competitive performance with as few as two iterations (layers). This demonstrates that the parameterization of temperature is more effective than employing a static parameterization.

Moreover, the hidden dimension $d$ significantly affects the expressiveness of our model. When the hidden dimension $d$ is small, the expressiveness is rather limited. As evidenced in Figure 4(b) when $d = 1$, the performance of NGA is comparable to that of traditional method. However, as $d$ increases, the performance steadily improves until it saturates at higher dimensions. This further validates the effectiveness of the temperature parameterization in NGA in accordance with Proposition 2. Based on this analysis, we set $m = 4$ and $d = 32$ in all experiments to strike a balance between performance and computational efficiency.

To broaden the scope and potential impact of our work, we have conducted additional experiments on a subset of QAPLIB (Burkard et al., 1997) instances across different categories in Appendix G.5, where our method exhibits competitive performance compared with baseline methods.

## 7 CONCLUSION

This study introduces Neural Graduated Assignment (NGA), a novel approach to the Maximum Common Edge Subgraph (MCES) problem, of which the scalability remains a challenge for traditional methods. Drawing inspirations from annealing mechanism in statistical physics, NGA employs a neural architecture to iteratively refine the assignment process, leveraging high-dimensional learnable parameters to improve computational efficiency, scalability, and adaptive problem-solving. Empirical results show that NGA significantly outperforms prior methods in both runtime and accuracy, demonstrating strong performance across tasks like graph similarity and retrieval. The introduction of this versatile method offers substantial contributions to the understanding and resolution of assignment problems, suggesting pathways for future exploration and innovation in related areas.

ACKNOWLEDGEMENTS

This work was supported by the National Key R&D Program of China under grant 2022YFA1003900.

ETHICS AND REPRODUCIBILITY STATEMENT

To ensure the reproducibility of our research, we provide a comprehensive set of resources. Code is open-sourced at `https://github.com/LOGO-CUHKSZ/NGA`. A detailed README file offers step-by-step guidance for setting up the computational environment and reproducing the experiments described in this paper.

This research focuses on methodological and technical contributions. It does not involve human subjects, personal data, or sensitive information. We are not aware of any direct ethical concerns arising from this work.

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

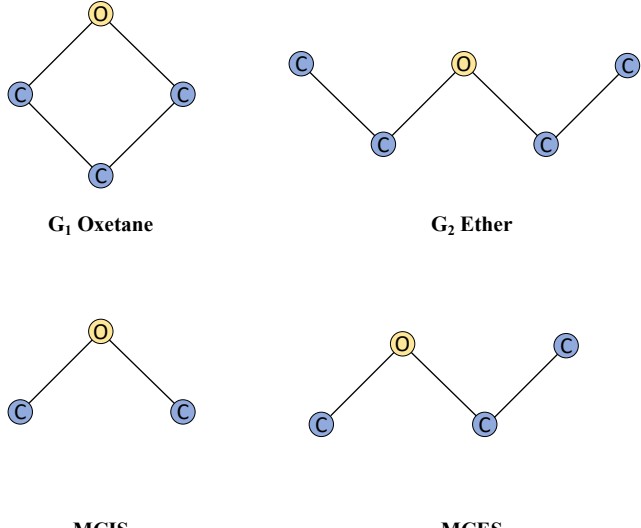

**G₁ Oxetane**     **G₂ Ether**

MCIS     MCES

Figure 5: The maximum common induced subgraph (MCIS) and the maximum common edge subgraph (MCES) of two labeled graphs.

## A   MORE BACKGROUND

**Maximum Common Subgraph.**   The maximum common subgraph (MCS) problem (Bunke & Shearer, 1998) is a fundamental topic in graph theory with significant implications in various real-world applications. Given two graphs, the goal of the MCS problem is to find the largest subgraph that is isomorphic to a subgraph of both input graphs. The MCS problem inherently captures the degree of similarity between two graphs, is domain-independent, and therefore finds extensive applications across various fields, such as substructure similarity search in databases (Yan et al., 2005), source code analysis (Djoko et al., 1997) and computer vision (Cook & Holder, 1993). The MCS problem can be broadly categorized into two variants: the maximum common induced subgraph (MCIS) and the maximum common edge subgraph (MCES). The key distinction between these variants lies in the structural constraints: MCIS focuses on finding a subgraph that preserves both vertex and edge connectivity, whereas MCES concentrates on maximizing common edge structure without enforcing vertex-induced constraints (Ndiaye & Solnon, 2011). An example of MCIS and MCES is shown in Figure 5. This paper focuses on the MCES problem, which is particularly applicable to practical tasks in bioinformatics, such as finding similar substructures in compounds with similar properties (Ehrlich & Rarey, 2011), where the edge-based relationships often carry significant biological meaning.

**Graph Similarity.**   A key challenge in pharmaceutical research involving small molecules is organizing individual compounds into structurally related families or clusters. Manually sorting through large databases can be labor-intensive, which is why automated methods are frequently employed. These automated clustering techniques require a similarity metric for pairwise structure comparison and a clustering algorithm to categorize compounds into related groups.

**Graph Retrieval.**   Graph retrieval involves identifying and ranking target graphs based on their similarity to a given query graph $G_q$ (Roy et al., 2022b;a). The task can be framed as a ranking problem, where a graph retrieval system assigns similarity scores or distance metrics to target graphs relative to $G_q$. The process typically consists of two steps: first, calculating a similarity score for each target graph $G_t$; and second, ranking the target graphs in decreasing order of similarity. The quality of a graph retrieval algorithm depends on its ability to place the most relevant graphs at the top of the ranking. A key challenge in this domain is the design of effective scoring functions. In this work, we adopt the Johnson similarity as scoring function, where the target graph with the highest similarity to $G_q$ is deemed the most relevant and ranked highest.

**Graph Neural Networks.** Graph Neural Networks (GNNs) are a class of deep learning models specifically designed to process data represented as graphs. By leveraging the graph structure, GNNs can capture complex dependencies and interactions, enabling them to learn rich representations of nodes and entire graphs. These models utilize message passing mechanisms to iteratively aggregate information from neighboring nodes, ultimately allowing them to excel in various applications. As GNNs continue to evolve, they offer powerful tools for tackling challenges in machine learning and artificial intelligence across diverse domains (Hamilton et al., 2017; Ying et al., 2024; Zhao et al., 2025). Given an input graph, typical GNNs compute node embeddings $\boldsymbol{h}_u^{(t)}, \forall u \in \mathcal{V}$ with $T$ layers of iterative message passing (Gilmer et al., 2017):

$$\boldsymbol{h}_u^{(t+1)} = \psi \left( \boldsymbol{h}_u^{(t)}, \sum_{v \in \mathcal{N}_u} \boldsymbol{h}_v^{(t)} \cdot \phi(\boldsymbol{e}_{uv}) \right), \tag{12}$$

for each $t \in [0, T-1]$, where $\mathcal{N}_u = \{v \in \mathcal{V} | (u,v) \in \mathcal{E}\}$, while $\psi$ and $\phi$ are neural networks, e.g. implemented using multilayer perceptrons (MLPs).

**Graduated Assignment.** The Graduated Assignment (GA) is a classical optimization method rooted in principles of statistical physics and commonly applied to assignment problems such as graph matching (Gold & Rangarajan, 1996; Cho et al., 2010). GA operates by iteratively updating a soft assignment matrix, which represents the probabilistic relationship between elements of two sets (e.g., nodes, edges) that are being matched. One should note that GA is not a neural method and only consists of forward updates.

## B    ASSOCIATION COMMON GRAPH EXPLANATION

**Proposition 1.** *Denote the node set of $G_1 \lozenge G_2$ as $\mathcal{V}_\lozenge = \{(u,v) | u \in \mathcal{V}(G_1), v \in \mathcal{V}(G_2)\}$. Consider a set of subgraphs of $G_1 \lozenge G_2$ where each node of $G_1$ and $G_2$ is selected at most once, i.e. any two nodes $(u_i, v_i)$ and $(u_j, v_j)$ in this set satisfy $u_i \neq u_j$ and $v_i \neq v_j$. **Any subgraph in this set is a valid common subgraph of $G_1$ and $G_2$, and finding the MCES of two graphs is reduced to finding the largest subgraph in this set.***

*Proof.* Denote the node set of $G_1 \lozenge G_2$ as $\mathcal{V}_\lozenge = \{(u_i, v_j) \mid u_i \in \mathcal{V}(G_1), v_j \in \mathcal{V}(G_2)\}$. In every common subgraph $G'$ of $G_1$ and $G_2$, each vertex is isomorphic to a distinct vertex in $\mathcal{V}(G_1)$ and $\mathcal{V}(G_2)$. If the subgraph of the ACG we select contains two vertices in the same row, it implies that a vertex in $G_2$ corresponds to two vertices in $G_1$, which is not valid. To prevent duplication in the mapping between the two graphs, we define a subgraph of the association graph as valid if and only if it satisfies the following conditions:

$$\begin{cases} \mathcal{V}(G) \subseteq \mathcal{V}_\lozenge, \\ \mathcal{E}(G) \subseteq \mathcal{E}_\lozenge, \\ |\mathcal{V}(G)| = \min\{|\mathcal{V}(G_1)|, |\mathcal{V}(G_2)|\}, \\ \forall (u_i, v_j), (u_k, v_l) \in \mathcal{V}(G), \quad i \neq k, \ j \neq l \end{cases} \tag{13}$$

We aim to show that each valid subgraph $G$ can be transformed into a common subgraph $G'$ of $G_1$ and $G_2$, satisfying $|\mathcal{E}(G)| = |\mathcal{E}(G')|$. In this way, finding the MCES of two graphs is reduced to finding the largest subgraph in this set. Let $\mathcal{G}_1$ denote the set of all valid subgraphs of $G_1 \lozenge G_2$, and let $\mathcal{G}_2$ denote the set of all common subgraphs of $G_1$ and $G_2$. We claim that there exists a surjective function $f : \mathcal{G}_1 \to \mathcal{G}_2$ that transforms $G$ into $G'$, such that $|\mathcal{E}(G)| = |\mathcal{E}(f(G))|$. If this claim holds, each subgraph of the association graph $G_1 \lozenge G_2$ corresponds to a common subgraph of $G_1$ and $G_2$ without omitting any common subgraph. Consequently, the subgraph with the largest number of edges corresponds to the MCES. Here we give the construction of $f$, and the proof of the surjection of function $f$. The edge set of $G$ is given by $\mathcal{E} = \{((u_i, v_i), (u_j, v_j)) \mid ((u_i, v_i), (u_j, v_j)) \in \mathcal{E}_\lozenge\}$. From this edge set, we restore all the edge information in the original graph $G_1$, denoted as $\mathcal{E}' = \{(u_i, u_j) \mid ((u_i, v_i), (u_j, v_j)) \in \mathcal{E}\} \subseteq \mathcal{E}(G_1)$. The corresponding vertex set is naturally defined as $\mathcal{V}' = \{u_k \mid \exists u_l, (u_k, u_l) \in \mathcal{E}'\}$. Thus, the subgraph $G' = \{\mathcal{E}', \mathcal{V}'\} \subseteq G_1$ is uniquely determined, and we define $f(G) = G'$. Similarly, we can construct $G'' = \{\mathcal{E}'' \subseteq \mathcal{E}(G_2), \mathcal{V}'' \subseteq \mathcal{V}(G_2)\} \subseteq G_2$. Since the matching pair in $\mathcal{E}$ is unique, $G'$ is isomorphic to $G''$. Hence, $G' = f(G)$ is the desired common subgraph of $G_1$ and $G_2$. Furthermore, it is straightforward to verify that $|\mathcal{E}(G)| = |\mathcal{E}(f(G))|$.

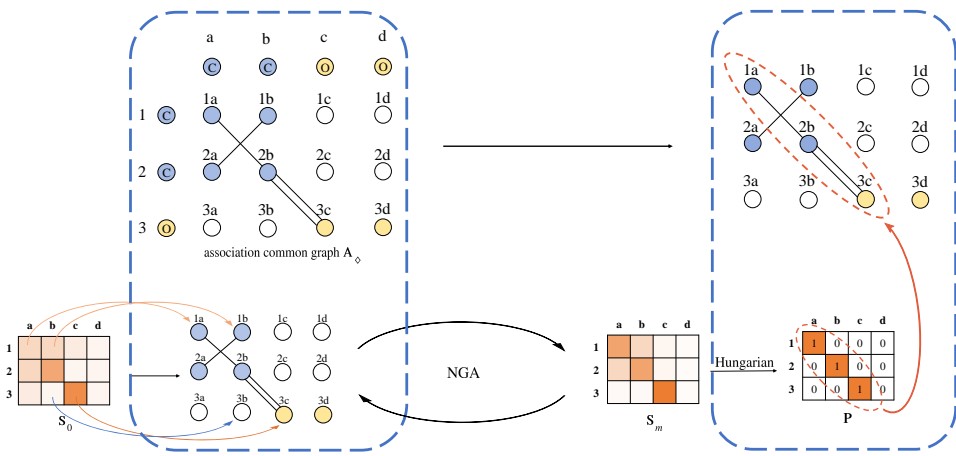

Figure 6: An illustration of the ACG. Each value in the soft assignment $\mathbf{S}$ or hard assignment $\mathbf{P}$ corresponds to a node in the ACG $\mathcal{A}_\diamond$ and thus can be considered as node features in $\mathcal{A}_\diamond$.

Next, we represent the node information of a valid subgraph $G$ of $G_1 \diamond G_2$ using an assignment matrix. Let $\mathbf{P} = (\mathbf{P}_{ij})$, where $\mathbf{P}_{ij} = 1$ if and only if $(u_i, v_j) \in \mathcal{V}(G)$. Since the association graph is already known, the assignment matrix $\mathbf{P}$ is sufficient to restore the valid subgraph. Thus, from this point forward, we only need to work with the assignment matrix. Any assignment can be be mapped to a common subgraph, thereby enhancing interpretability of our formulation of MCES. Consider an example in Figure 6, where NGA outputs an assignment matrix with entries $\mathbf{P}_{1a} = \mathbf{P}_{2b} = \mathbf{P}_{3c} = 1$. In this case, the assignment represents a valid subgraph $G$ with vertices $\mathcal{V}(G) = \{1a, 2b, 3c\}$. Using the association graph, it is straightforward to derive the unique corresponding subgraph $G$ and the original common subgraph $G'$ by applying the function $f$. Thus, instead of working directly with $G'$ or $G$, we can represent the subgraph using only the assignment matrix.

$\square$

## C EXPERIMENTAL SETUP

### C.1 DATASETS

Table 4: The statistics of datasets.

|  | #graphs | #nodes | #edges |
|---|---|---|---|
| AIDS | 2000 | $\sim 15.7$ | $\sim 16.2$ |
| MCF-7 | 27770 | $\sim 26.4$ | $\sim 28.5$ |
| MOLHIV | 41127 | $\sim 25.5$ | $\sim 27.5$ |

The key statistics of datasets used in this paper are summarized in Table 4. The solution time distributions of three datasets are illustrated in Figure 7.

### C.2 DISCUSSION OF BASELINE METHODS

For our comparison, we select baseline methods that are directly or indirectly related to solving the MCES problem. RASCAL and FMCS are the most relevant baselines, as they are specifically designed to solve MCES. While some methods, such as Mcsplit, are primarily developed for the MCIS problem, they are still capable of generating common subgraphs. According to the definition of MCES, the number of edges in the common subgraph of an MCIS will always be less than or equal to that of an MCES, i.e., $|\mathcal{E}(\mathrm{MCIS}(G_1, G_2))| \leq |\mathcal{E}(\mathrm{MCES}(G_1, G_2))|$. As such, MCIS solvers are also valid comparisons. However, some methods are not suitable for serving as compared baselines. Some studies, such as (Bahiense et al., 2012), disregard edge labels that are essential for molecular graphs.

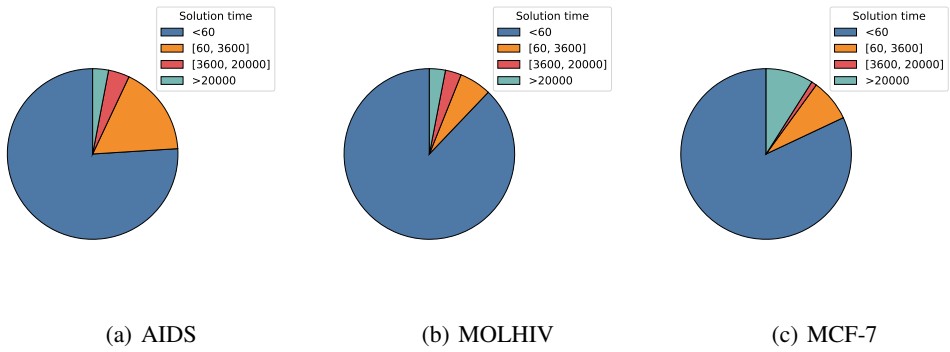

(a) AIDS      (b) MOLHIV      (c) MCF-7

Figure 7: Solution time distributions on three datasets

XMCS and INFMCS, on the other hand, only implicitly infer the size of the MCS for downstream tasks without providing explicit MCS structures. As a result, these methods are used only as baselines in our graph similarity computation and graph retrieval experiments.

Besides, we provide time complexity comparison between baseline methods and ours in this section. For notation simplicity, we assume that both the input graphs have $N$ nodes and $E$ edges. The construction of the ACG requires enumerating all edge pairs between the two input graphs $G_1$ and $G_2$, leading to a time complexity of $O(E^2)$. In practice, however, this cost is very limited because molecular graphs are typically sparse — each atom usually has only a few neighbors (typically up to 4) due to chemical valency constraints and thus $E^2 \ll N^2$. Besides, an edge in the ACG exists only when the corresponding pair of edges in $G_1$ and $G_2$ satisfy the compatibility constraints. Consequently, the actual number of edges $M$ in ACG satisfies $M < E^2 \ll N^2$. The space complexity of constructing the ACG is $O(M)$. To handle this efficiently, we can store the ACG using sparse data structures, such as sparse tensors, which allow memory usage to scale with $O(M)$ rather than with the dense upper bound. This ensures that the space overhead remains manageable even for relatively large graphs.

The initial assignments are obtained by GNN with a complexity of $\mathcal{O}(N^2 d)$. Therefore, the overall complexity is $\mathcal{O}(M + N^2)$. The complexity of NGA iteration is $\mathcal{O}(mk(M + N^2))$, where $m$ is the number of iterations and $k$ is the number of epochs in training. Overall, the time complexity of our method is $\mathcal{O}(mk(M + N^2))$. GA has the same iteration process as our method, which has a complexity of $\mathcal{O}(m(M + N^2))$. NGM has the same complexity as our method, which is $\mathcal{O}(mk(M + N^2))$. The rest baseline methods, including RASCAL, FMCS and Mcsplit have a complexity of approximately $\mathcal{O}(2^N)$ due to the NP-complete nature of the MCES problem. Compared with these baseline methods, our proposed method provides efficient approximations of the solution in polynomial time without relying on exhaustive exploration of the solution space.

## D   CONVERGENCE ANALYSIS

In this section, we first analyze the convergence behavior under our learnable temperature parameterization, and then prove the advantages of product parameterization.

### D.1   CONVERGENCE BEHAVIOR UNDER LEARNABLE PARAMETERS

The quadratic assignment objective function in Eq. (1) is formulated as follows (Rangarajan et al., 1996a) with regularizations:

$$
\begin{aligned}
\mathbb{E}(\mathbf{S}, \mu, \nu) = &-\frac{1}{2}\mathrm{vec}(\mathbf{S})^\top \mathcal{A}_\diamond \mathrm{vec}(\mathbf{S}) + \mu^\top(\mathbf{S}\mathbf{1}_{n_2} - \mathbf{1}_{n_1}) \\
&+ \nu(\mathbf{S}^\top \mathbf{1}_{n_1} - \mathbf{1}_{n_2}) - \frac{\gamma}{2}\mathrm{vec}(\mathbf{S})^\top \mathrm{vec}(\mathbf{S}) \\
&+ \frac{1}{\beta}\sum \mathbf{S}\log \mathbf{S},
\end{aligned}
\tag{14}
$$

where $\mathbb{E}$ is the energy function, $\mu$ and $\nu$ are Lagrange parameters for constraint satisfaction, $\gamma$ is a parameter for the self-amplification term, and $\beta$ is a deterministic annealing control parameter.

We analyze the convergence behavior at each iteration layer and temporarily omit the layer index $(i)$ for simplicity and denote $\beta_l = \mathbf{W}_1^\top \mathbf{W}_2$ as the learnable parameter. In the first part of the proof, we show that a proper choice of parameter $\gamma$, is guaranteed to decrease the energy function. The energy function in Eq. (14) can be simplified by collecting together all quadratic terms in $\mathbf{S}$, i.e. we define

$$\mathcal{A}_\Diamond^{(\gamma)} = \mathcal{A}_\Diamond + \gamma \mathbf{I}. \tag{15}$$

Consider the following algebraic transformation:

$$-\frac{\mathbf{X}^\top \mathbf{X}}{2} \to \min_\sigma (-\mathbf{X}^\top \sigma + \frac{\sigma^\top \sigma}{2}) \tag{16}$$

and we can convert the quadratic form into a simpler, more tractable linear form:

$$\begin{aligned}
\mathbb{E}(\mathbf{S}, \mu, \nu, \sigma) = &-\text{vec}(\mathbf{S})^\top \mathcal{A}_\Diamond^{(\gamma)} \sigma + \frac{1}{2} \sigma^\top \mathcal{A}_\Diamond^{(\gamma)} \sigma \\
&+ \mu^\top (\mathbf{S}\mathbf{1}_{n_2} - \mathbf{1}_{n_1}) + \nu(\mathbf{S}^\top \mathbf{1}_{n_1} - \mathbf{1}_{n_2}) \\
&+ \frac{1}{\beta_l} \sum \mathbf{S} \log \mathbf{S}
\end{aligned} \tag{17}$$

We use $deg_i$ to represent the degree of node $i$ and $deg_{\max} = \max_{i \in \mathcal{V}_\Diamond} deg_i$. By setting $\gamma = deg_{\max} + \epsilon$, where $\epsilon > 0$ is a small quantity, the following equation holds for any nonzero vector $\mathbf{x}$:

$$\begin{aligned}
&\mathbf{x}^\top \mathcal{A}_\Diamond^{(\gamma)} \mathbf{x} \\
&= \mathbf{x}^\top \mathcal{A}_\Diamond \mathbf{x} + \mathbf{x}^\top \text{diag}(\gamma) \mathbf{x} \\
&= \sum_{j=1}^n (\sum_{i=1}^n \mathbf{x}_i a_{ij}) \mathbf{x}_j + \sum_{i=1}^n \mathbf{x}_i^2 (deg_i + deg_{max} - deg_i + \epsilon) \\
&= \frac{1}{2} (\sum_{i=1}^n deg_i \mathbf{x}_i^2 + 2 \sum_{i,j=1}^n \mathbf{x}_i \mathbf{x}_j a_{ij} + \sum_{j=1}^n deg_j \mathbf{x}_j^2) \\
&+ \sum_{i=1}^n \mathbf{x}_i^2 (deg_{max} - deg_i + \epsilon) \\
&= \frac{1}{2} \sum_{i,j=1}^n a_{ij} (\mathbf{x}_i + \mathbf{x}_j)^2 + \sum_{i=1}^n \mathbf{x}_i^2 (deg_{max} - d_i + \epsilon) \\
&> 0
\end{aligned} \tag{18}$$

which proves that $\mathcal{A}_\Diamond^{(\gamma)}$ is positive definite.

Extremizing Eq. (17) with respect to $\sigma$, we have

$$\mathcal{A}_\Diamond^{(\gamma)} \text{vec}(\mathbf{S}) = \mathcal{A}_\Diamond^{(\gamma)} \sigma \quad \Rightarrow \quad \sigma = \text{vec}(\mathbf{S}) \tag{19}$$

is a minimum, which means that setting $\sigma = \text{vec}(\mathbf{S})$ is guaranteed to decrease the energy function.

In the second part of the proof, we show how the energy function in Eq. (17) increases or decreases according to $\beta_l$. In the first part of the proof, we set $\sigma = \text{vec}(\mathbf{S})$, referring $\sigma$ as the original value of $\text{vec}(\mathbf{S})$. To achieve convergence, we need to show that $\mathbb{E}(\sigma, \sigma) \geq \mathbb{E}(\text{vec}(\mathbf{S}), \sigma)$ in Eq. (17). The $\mathbf{S}$ here is the new doubly stochastic matrix gained after executing the Sigmoid and Sinkhorn operator. After each iteration, the new matrix $\mathbf{S}$ always reduces the value of the objective function.

Minimizing Eq. (17) with respect to $\mathbf{S}_{ai}$, we get

$$\frac{1}{\beta_l} \log \mathbf{S}_{ai} = [\mathcal{A}_\Diamond^{(\gamma)} \sigma]_{ai} - (\mu_a + \nu_i) - \frac{1}{\beta_l} \tag{20}$$

From Eq. (20), we get the summation

$$
\frac{1}{\beta_l} \sum \mathbf{S}_{ai} \log \mathbf{S}_{ai} =
$$
$$
\mathrm{vec}(\mathbf{S})^T \mathcal{A}_\Diamond^{(\gamma)} \sigma - \mu^T \mathrm{vec}(\mathbf{S}) \mathbf{1}_{n_2}
$$
$$
+ \nu \mathrm{vec}(\mathbf{S})^T \mathbf{1}_{n_1} - \frac{1}{\beta_l} \sum \mathbf{S}_{ai}
$$

(21)

and

$$
\frac{1}{\beta_l} \sum \sigma_{ai} \log \mathbf{S}_{ai} =
$$
$$
\sigma^T \mathcal{A}_\Diamond^{(\gamma)} \sigma - \mu^T \sigma \mathbf{1}_{n_2} + \nu \sigma^T \mathbf{1}_{n_1} - \frac{1}{\beta_l} \sum \sigma_{ai}
$$

(22)

From Eq. (21) and Eq. (22), we get

$$
\mathbb{E}(\sigma, \sigma) - \mathbb{E}(\mathrm{vec}(\mathbf{S}), \sigma) =
$$
$$
-\sigma^T \mathcal{A}_\Diamond^{(\gamma)} \sigma - (-\mathrm{vec}(\mathbf{S})^T \mathcal{A}_\Diamond^{(\gamma)} \sigma)
$$
$$
+ \sum \frac{\sigma_{ai}}{\beta_l} \log \sigma_{ai} - \sum \frac{\mathbf{S}_{ai}}{\beta_l} \log \mathbf{S}_{ai}
$$
$$
= \frac{1}{\beta_l} \sum \sigma_{ai} \log \frac{\sigma_{ai}}{\mathbf{S}_{ai}}
$$

(23)

By the non-negativity of the Kullback-Leibler measure, $\mathbb{E}(\sigma, \sigma) - \mathbb{E}(\mathrm{vec}(\mathbf{S}), \sigma) = \frac{1}{\beta_l} \sum \sigma \log \frac{\sigma}{\mathbf{S}} \geq 0$ when $\beta_l > 0$ and vice versa. The doubly stochastic property of $\mathbf{S}$ and $\sigma$ ensures that the Lagrange parameters can be eliminated from the energy function Eq. (17). The new matrix $\mathbf{S}$ gained after setting $\sigma = \mathrm{vec}(\mathbf{S})$ guarantees the decrease of the energy function.

We distill the core of the proof to highlight the key aspects. At each temperature, the algorithm performs the following steps repeatedly until it reaches convergence.

**Step 1:** $\sigma \leftarrow \mathrm{vec}(\mathbf{S})$
**Step 2:**

**Step 2a:** $\hat{\mathbf{S}}_{ai} \leftarrow \exp(\beta_l \sum_{bj} (\mathcal{A}_\Diamond^{(\gamma)})_{ai,bj} \sigma_{bj})$

(24)

**Step 2b:** $\mathbf{S} \leftarrow \mathrm{Sinkhorn}(\hat{\mathbf{S}})$
Return to Step 1 until convergence.

Our proof shows that when a suitably constructed energy function is affect by the temperature parameter $\beta_l$ during both Step 1 and Step 2. This energy function corresponds to Eq. (17) without the terms involving the Lagrange parameters.

### D.2 PROOF OF FASTER CONVERGENCE FOR PRODUCT PARAMETERIZATION

**Proposition 2.** *Under typical gradient descent conditions where the gradient pushes $\beta_l$ in a consistent direction for multiple iterations, the product parameterization $\beta_l = \mathbf{W}_1^\top \mathbf{W}_2$ can adaptively adjust the learning rate and induces an accelerating effect on the magnitude of updates to $\beta_l$.*

*Proof.* Let $J(\mathbf{S}) = \mathrm{vec}(\mathbf{S})^\top \mathcal{A}_\Diamond \mathrm{vec}(\mathbf{S})$ and $g(t) = \frac{\partial J}{\partial \beta_l(t)}$ be the gradient of the loss at step $t$. For a direct scalar parameterization $\beta_l$ and learning rate $\eta$, the gradient update reads:

$$
\beta_l(t+1) = \beta_l(t) + \eta g(t).
$$

(25)

For the product parameterization, the update is:

$$
\mathbf{W}_1(t+1) = \mathbf{W}_1(t) + \eta g(t) \mathbf{W}_2(t), \quad \mathbf{W}_2(t+1) = \mathbf{W}_2(t) + \eta g(t) \mathbf{W}_1(t)
$$

(26)

Let's examine how $\beta_l$ changes over time. After one update,

$$
\begin{aligned}
\beta_l(t+1) &= \mathbf{W}_1(t+1)^\top \mathbf{W}_2(t+1) \\
&= (\mathbf{W}_1(t) + \eta g(t)\mathbf{W}_2(t))^\top (\mathbf{W}_2(t) + \eta g(t)\mathbf{W}_1(t)) \\
&= \mathbf{W}_1(t)^\top \mathbf{W}_2(t) + \eta g(t)\mathbf{W}_1(t)^\top \mathbf{W}_1(t) + \eta g(t)\mathbf{W}_2(t)^\top \mathbf{W}_2(t) + \eta^2 g(t)^2 \mathbf{W}_2(t)^\top \mathbf{W}_1(t) \\
&= \beta_l(t) + \eta g(t)\left(\|\mathbf{W}_1(t)\|^2 + \|\mathbf{W}_2(t)\|^2\right) + \eta^2 g(t)^2 \beta_l(t)
\end{aligned}
$$
(27)

This is a nonlinear update equation with respect to the gradient $g(t)$, containing both the first-order term $\eta g(t)\left(\|\mathbf{W}_1(t)\|^2 + \|\mathbf{W}_2(t)\|^2\right)$ and the second-order term $\eta^2 g(t)^2 \beta_l(t)$. If we omit the higher order gradients, the term $\mathbf{W}_1(t)^\top \mathbf{W}_2(t)$ can adaptively adjust the learning rate.

Let's analyze the scenario where the gradient $g(t) > 0$ for several consecutive iterations, indicating that increasing $\beta_l$ would increase the loss. This is a common scenario in optimization where the algorithm consistently moves in a beneficial direction. For the direct scalar parameterization, the update is

$$
\beta_l(t+1) = \beta_l(t) + \eta |g(t)|.
$$
(28)

After $k$ iterations with approximately constant gradient magnitude $|g|$, we get

$$
\beta_l(t+k) \approx \beta_l(t) + k\eta|g|.
$$
(29)

Let $\mathbf{W}_1(0) = \mathbf{W}_0$, $\mathbf{W}_2(0) = \mathbf{U}_0$ as the initialization state, we can derive:

$$
\begin{aligned}
\mathbf{W}_1(k) &\approx \frac{(\mathbf{W}_0 + \mathbf{U}_0)}{2}(1+\eta|g|)^k + \frac{(\mathbf{W}_0 - \mathbf{U}_0)}{2}(1-\eta|g|)^k \\
\mathbf{W}_2(k) &\approx \frac{(\mathbf{W}_0 + \mathbf{U}_0)}{2}(1+\eta|g|)^k - \frac{(\mathbf{W}_0 - \mathbf{U}_0)}{2}(1-\eta|g|)^k.
\end{aligned}
$$
(30)

Thus, the first-order term becomes:

$$
\begin{aligned}
&\eta|g|\left(\|\mathbf{W}_1(t)\|^2 + \|\mathbf{W}_2(t)\|^2\right) \\
&= \frac{\eta|g|}{2}\big[\left(\|\mathbf{W}_0\|^2 + \|\mathbf{U}_0\|^2\right)\left((1+\eta|g|)^{2k} + (1-\eta|g|)^{2k}\right) \\
&\quad + 2\left(\mathbf{W}_0^\top \mathbf{U}_0\right)\left((1+\eta|g|)^{2k} - (1-\eta|g|)^{2k}\right)\big],
\end{aligned}
$$
(31)

which grows exponentially with the iteration count $k$. Therefore, the product parameterization leads to accelerating updates to $\beta_l$. In contrast, the direct scalar parameterization has a constant update magnitude $\eta|g|$ regardless of the iteration count. Aside from this, this reparameterization naturally introduces bounds on the magnitude of $\beta_l$: $|\beta_l| \leq \|\mathbf{W}_1^{(l)}\|\|\mathbf{W}_2^{(l)}\| \leq (\frac{\|\mathbf{W}_1^{(l)}\|^2 + \|\mathbf{W}_2^{(l)}\|^2}{2})^2$. $\qquad\square$

Moreover, this proposition is supported by empirical evidence. Specifically, we provide the loss curves in Figure 13. Compared to parameterizing the temperature as a learnable scalar, our method exhibits faster convergence. Within the same training time, this results in improved performance. As shown in Table 5 from Appendix F, the product parameterization (NGA) outperforms the scalar version (NGA-Scalar) by approximately 15%. These results highlight the effectiveness of our method compared to directly parameterizing $\beta_l$ as a learnable scalar.

Recent works in the implicit-bias literature have shown that reparameterized models—particularly those involving quadratic or low-rank factorizations—can induce non-trivial optimization dynamics that resemble adaptive or mirror-descent–like updates. For example, Li et al. (2022) and Jacobs et al. (2025) demonstrate that such reparameterizations implicitly modify the effective geometry of gradient descent, leading to adaptive learning-rate behaviors without explicit scheduling. Related work on sparsity-inducing training (Jacobs & Burkholz, 2025; Kolb et al., 2025) further illustrates how factored parameterizations can shape convergence trajectories and amplify or damp particular update directions.

Our product parameterization of the temperature term shares conceptual similarities with these findings. These perspectives provide additional insight into why our reparameterizations can yield favorable optimization behavior in our setting.

# E   BEHAVIOR AT LOCAL OPTIMA

Before proving Theorem 1, we first conclude the following Lemma.

**Lemma 1.** *For a local optimum* $\mathbf{S}^{(l)}$*, and for* $|\beta_l|$ *sufficiently small, the following property holds:*

$$\mathbf{S}_{ij}^{(l+1)} = \frac{1}{n}[1 + \beta_l(h_{ij} - \bar{h}) + O(|\beta_l|^2)] \tag{32}$$

*where* $h_{ij} = [\mathcal{A}_\Diamond \text{vec}(\mathbf{S}^{(l)})]_{ij}$ *and* $\bar{h}$ *is the mean of* $h_{ij}$ *over all* $i$ *and* $j$*.*

*Proof.* First, we can characterize the pre-Sinkhorn matrix $\mathbf{M}^{(l)}$ through its log-space representation:

$$\ln \mathbf{M}_{ij}^{(l)} = \beta_l h_{ij} \tag{33}$$

By the Sinkhorn-Knopp theorem (Sinkhorn & Knopp, 1967), there exist diagonal matrices $\mathbf{D}_r$ and $\mathbf{D}_c$ with positive entries such that:

$$\mathbf{S}^{(l+1)} = \mathbf{D}_r \mathbf{M}^{(l)} \mathbf{D}_c \tag{34}$$

is doubly stochastic. Taking logarithm we have:

$$\ln \mathbf{S}_{ij}^{(l+1)} = \ln \mathbf{D}_r^{(i)} + \beta_l h_{ij} + \ln \mathbf{D}_c^{(j)} \tag{35}$$

Let's define the potentials $u_i = \ln \mathbf{D}_r^{(i)}$ and $v_j = \ln \mathbf{D}_c^{(j)}$. These satisfy:

$$\exp(u_i + \beta_l h_{ij} + v_j) = \mathbf{S}_{ij}^{(l+1)} \tag{36}$$

$$\begin{aligned}
\sum_j \mathbf{S}_{ij}^{(l+1)} = \sum_j \exp(u_i + \beta_l h_{ij} + v_j) = 1 \\
\sum_i \mathbf{S}_{ij}^{(l+1)} = \sum_i \exp(u_i + \beta_l h_{ij} + v_j) = 1
\end{aligned} \tag{37}$$

When $\beta_l = 0$, the Sinkhorn algorithm has a trivial solution:

$$\mathbf{S}_{ij}^{(l+1)} = \frac{1}{n}, \quad u_i^{(0)} = v_j^{(0)} = -\frac{1}{2}\ln n \tag{38}$$

For small $|\beta_l|$, we can view the solution as a perturbation around this base solution. By the implicit function theorem, if the Sinkhorn algorithm converges (guaranteed by our assumptions) and $|\beta_l|$ is sufficiently small, the functions involved are smooth.

Let's write the system of equations that defines our problem. For the doubly stochastic constraints (Eq. (37)):

$$F_i(u, v, \beta_l) = \sum_j \exp(u_i + \beta_l h_{ij} + v_j) - 1 = 0 \quad \forall i \tag{39}$$

$$G_j(u, v, \beta_l) = \sum_i \exp(u_i + \beta_l h_{ij} + v_j) - 1 = 0 \quad \forall j \tag{40}$$

For this system, the smoothness can be verified by checking the insingularity of the Jacobian matrix. Therefore, by the implicit function theorem, there exist unique (up to addition of constants) functions $u_i(\beta_l)$ and $v_j(\beta_l)$ in a neighborhood of $\beta_l = 0$ that satisfy our constraints and are continuously differentiable. Since these functions are differentiable at $\beta_l = 0$, we can write their Taylor expansions:

$$\begin{aligned}
u_i(\beta_l) = u_i^{(0)} + \beta_l u_i^{(1)} + O(|\beta_l|^2) \\
v_j(\beta_l) = v_j^{(0)} + \beta_l v_j^{(1)} + O(|\beta_l|^2)
\end{aligned} \tag{41}$$

where $u_i^{(1)} = \frac{du_i}{d\beta_l}|_{\beta_l=0}$ and $v_j^{(1)} = \frac{dv_j}{d\beta_l}|_{\beta_l=0}$. Then the potentials $u_i$ and $v_j$ are analytic functions of $\beta_l$ near 0. To determine $u_i^{(1)}$ and $v_j^{(1)}$, we use the row and column sum constraints:

$$\sum_j \exp(u_i + \beta_l h_{ij} + v_j) = 1 \tag{42}$$

Substituting the expansions (as when $\beta_l = 0$, we have $u_i^{(0)} = v_j^{(0)} = -\frac{1}{2} \ln n$)

$$
\begin{aligned}
u_i &= -\frac{1}{2} \ln n + \beta_l u_i^{(1)} + O(|\beta_l|^2) \\
v_j &= -\frac{1}{2} \ln n + \beta_l v_j^{(1)} + O(|\beta_l|^2)
\end{aligned}
\tag{43}
$$

into the numerator of $\mathbf{S}_{ij}^{(l+1)}$:

$$
\begin{aligned}
&\exp(\beta_l h_{ij} + u_i + v_j) \\
&= \exp(\beta_l h_{ij} - \ln n + \beta_l(u_i^{(1)} + v_j^{(1)}) + O(|\beta_l|^2)) \\
&= \frac{1}{n} \exp(\beta_l(h_{ij} + u_i^{(1)} + v_j^{(1)}) + O(|\beta_l|^2)) \\
&= \frac{1}{n}(1 + \beta_l(h_{ij} + u_i^{(1)} + v_j^{(1)}) + O(|\beta_l|^2))
\end{aligned}
\tag{44}
$$

Inside the exponential:

$$
\begin{aligned}
&u_i + \beta_l h_{ij} + v_j \\
&= (-\frac{1}{2} \ln n + \beta_l u_i^{(1)} + O(|\beta_l|^2)) + \beta_l h_{ij} + (-\frac{1}{2} \ln n + \beta_l v_j^{(1)} + O(|\beta_l|^2)) \\
&= -\ln n + \beta_l(u_i^{(1)} + h_{ij} + v_j^{(1)}) + O(|\beta_l|^2)
\end{aligned}
\tag{45}
$$

Therefore:

$$
\begin{aligned}
&\exp(u_i + \beta_l h_{ij} + v_j) \\
&= \exp(-\ln n + \beta_l(u_i^{(1)} + h_{ij} + v_j^{(1)}) + O(|\beta_l|^2)) \\
&= \frac{1}{n} \exp(\beta_l(u_i^{(1)} + h_{ij} + v_j^{(1)}) + O(|\beta_l|^2))
\end{aligned}
\tag{46}
$$

For small $|\beta_l|$, using Taylor expansion of exponential:

$$
\begin{aligned}
&\exp(\beta_l(u_i^{(1)} + h_{ij} + v_j^{(1)}) + O(|\beta_l|^2)) \\
&= 1 + \beta_l(u_i^{(1)} + h_{ij} + v_j^{(1)}) + O(|\beta_l|^2)
\end{aligned}
\tag{47}
$$

Then we have:

$$
\begin{aligned}
&\sum_j \exp(u_i + \beta_l h_{ij} + v_j) \\
&= \sum_j \frac{1}{n}(1 + \beta_l(u_i^{(1)} + h_{ij} + v_j^{(1)}) + O(|\beta_l|^2)) \\
&= \frac{1}{n} \sum_j (1 + \beta_l(u_i^{(1)} + h_{ij} + v_j^{(1)}) + O(|\beta_l|^2)) = 1
\end{aligned}
\tag{48}
$$

For this equation to be consistent for small $|\beta_l|$, we must have:

$$
\sum_j (u_i^{(1)} + h_{ij} + v_j^{(1)}) = 0, \qquad \sum_i (u_i^{(1)} + h_{ij} + v_j^{(1)}) = 0
\tag{49}
$$

These equations determine $u_i^{(1)}$ and $v_j^{(1)}$ up to an additive constant.

Next, we check how the overall behavior of $\mathbf{S}^{(l+1)}$ when $|\beta_l|$ is sufficiently small. For the assignment matrix:

$$
\mathbf{S}_{ij}^{(l+1)} = \frac{\exp(\beta_l h_{ij} + u_i + v_j)}{\sum_{k,m} \exp(\beta_l h_{km} + u_k + v_m)}
\tag{50}
$$

Substitute Eq. (43) into the numerator, we get the same formulation as Eq. (44). For $|\beta_l|$ sufficiently small, the subsequent Sinkhorn layer can converge very fast. Therefore, in the denominator:

$$
\begin{aligned}
\sum_{k,m} & \exp(\beta_l h_{km} + u_k + v_m) \\
&= \sum_{k,m} \frac{1}{n}(1 + \beta_l(h_{km} + u_k^{(1)} + v_m^{(1)}) + O(|\beta_l|^2)) \\
&= 1 + \beta_l(\frac{1}{n}\sum_{k,m} h_{km} + \sum_k u_k^{(1)} + \sum_m v_m^{(1)}) + O(|\beta_l|^2)
\end{aligned}
\tag{51}
$$

Let's define $\bar{h} = \frac{1}{n^2}\sum_{k,m} h_{km}$. Using Eq. (49) and the division formula $(1 + a)/(1 + b) = 1 + (a - b) + O((a - b)^2)$ for small $a, b$:

$$
\begin{aligned}
\mathbf{S}_{ij}^{(l+1)} &= \frac{\frac{1}{n}(1 + \beta_l(h_{ij} + u_i^{(1)} + v_j^{(1)}) + O(|\beta_l|^2))}{1 + \beta_l(n^2\bar{h} + \sum_k u_k^{(1)} + \sum_m v_m^{(1)}) + O(|\beta_l|^2)} \\
&= \frac{1}{n}[1 + \beta_l(h_{ij} + u_i^{(1)} + v_j^{(1)} - n^2\bar{h} - \sum_k u_k^{(1)} - \sum_m v_m^{(1)}) + O(|\beta_l|^2)] \\
&= \frac{1}{n}[1 + \beta_l(h_{ij} - \bar{h}) + O(|\beta_l|^2)]
\end{aligned}
\tag{52}
$$

where $\bar{h}$ is the mean of $h_{ij}$. $\qquad\square$

Having Lemma 1, we can further conclude the following theorem.

**Theorem 1.** *Given a local optimum $\mathbf{S}^{(l)}$, the change $\Delta J = J(\mathbf{S}^{(l+1)}) - J(\mathbf{S}^{(l)})$ satisfies*

$$
\Delta J = \frac{1}{2}(\sum_i \lambda_i - \sum_j \mu_j) + \beta_l \cdot \mathrm{Var}(h) + O(|\beta_l|^2),
\tag{9}
$$

*where $h_{ij} = [\mathcal{A}_\diamond \mathrm{vec}(\mathbf{S}^{(l)})]_{ij}$, $\mathrm{Var}(h)$ is the variance, $J(\mathbf{S}^{(l)}) = \mathrm{vec}(\mathbf{S}^{(l)})^\top \mathcal{A}_\diamond \mathrm{vec}(\mathbf{S}^{(l)})$ is the objective, $\lambda_i$ and $\mu_j$ are the Lagrange multipliers.*

*Proof.* Expand the change in objective value, we have:

$$
\begin{aligned}
\Delta J &= J(\mathbf{S}^{(l+1)}) - J(\mathbf{S}^{(l)}) \\
&= \mathrm{vec}(\mathbf{S}^{(l+1)} - \mathbf{S}^{(l)})^\top \mathcal{A}_\diamond \mathrm{vec}(\mathbf{S}^{(l)}) + \frac{1}{2}\mathrm{vec}(\mathbf{S}^{(l+1)} - \mathbf{S}^{(l)})^\top \mathcal{A}_\diamond \mathrm{vec}(\mathbf{S}^{(l+1)} - \mathbf{S}^{(l)})
\end{aligned}
\tag{53}
$$

From Lemma 1

$$
\mathbf{S}_{ij}^{(l+1)} - \mathbf{S}_{ij}^{(l)} = \frac{1}{n}[1 + \beta_l(h_{ij} - \bar{h}) + O(|\beta_l|^2)] - \mathbf{S}_{ij}^{(l)}
\tag{54}
$$

Let's analyze the first term in Eq. (53):

$$
\begin{aligned}
\mathrm{vec}(\mathbf{S}^{(l+1)} &- \mathbf{S}^{(l)})^\top \mathcal{A}_\diamond \mathrm{vec}(\mathbf{S}^{(l)}) \\
&= \sum_{i,j}(\frac{1}{n}[1 + \beta_l(h_{ij} - \bar{h}) + O(|\beta_l|^2)] - \mathbf{S}_{ij}^{(l)})h_{ij} \\
&= \sum_{i,j}[\frac{1}{n} - \mathbf{S}_{ij}^{(l)}]h_{ij} + \frac{\beta_l}{n}\sum_{i,j}(h_{ij} - \bar{h})h_{ij} + O(|\beta_l|^2)
\end{aligned}
\tag{55}
$$

Since $\mathrm{Var}(h) = \frac{1}{n^2}\sum_{i,j} h_{ij}^2 - \bar{h}^2$:

$$
\mathrm{vec}(\mathbf{S}^{(l+1)} - \mathbf{S}^{(l)})^\top \mathcal{A}_\diamond \mathrm{vec}(\mathbf{S}^{(l)}) = \sum_{i,j}[\frac{1}{n} - \mathbf{S}_{ij}^{(l)}]h_{ij} + \beta_l \cdot \mathrm{Var}(h) + O(|\beta_l|^2)
\tag{56}
$$

At the local optimum, by KKT conditions:

$$2h_{ij} - \lambda_i - \mu_j = 0 \tag{57}$$

for entries where $\mathbf{S}_{ij}^{(l)} > 0$. $\lambda_i$ and $\mu_j$ are Lagrange multipliers. Therefore,

$$
\begin{aligned}
\sum_{i,j}[\frac{1}{n} - \mathbf{S}_{ij}^{(l)}]h_{ij} &= \sum_{i,j}[\frac{1}{n} - \mathbf{S}_{ij}^{(l)}]\frac{\lambda_i + \mu_j}{2} \\
&= \frac{1}{2}(\sum_i \lambda_i - \sum_j \mu_j)
\end{aligned}
\tag{58}
$$

using the doubly stochastic constraints.

It is easy to show that the second-order term is $O(|\beta_l|^2)$. Therefore, summing the two terms we have

$$\Delta J = \frac{1}{2}(\sum_i \lambda_i - \sum_j \mu_j) + \beta_l \cdot \text{Var}(h) + O(|\beta_l|^2) \tag{59}$$

$\square$

## F  ABLATION STUDY

To better understand the contribution of each component in our framework, we conduct an ablation study by systematically removing or modifying key modules. This analysis highlights the effectiveness of individual design choices and demonstrates how each component contributes to the overall performance. The results are summarized in Table 5.

We first propose an ablated variant named NGA-w/o GNN, which removes GNN and instead employs a randomly initialized matrix for $\mathbf{S}^{(0)}$. Experimental results reveal that this simplified variant maintains strong performance in solving MCES, thereby demonstrating the crucial role and inherent effectiveness of our subsequent NGA refinement algorithm. While this variant achieves satisfactory results, the integration of GNN generates more optimized initial node assignments through learned representations. This enhanced initialization, when coupled with the NGA refinement process, translates to superior overall performance.

To demonstrate the effectiveness of the exploration and exploitation process of our proposed NGA, we tested a variant of our method where we use the Sigmoid function to restrict $\beta_l > 0$, denoted as NGA-Positive. This variant disables the exploration ability of our method. We observe that restricting $\beta_l$ to positive values significantly degrades performance, indicating that allowing negative temperatures is crucial for effective exploration and for enabling NGA to escape local minima.

To systematically evaluate the sensitivity of NGA to Gumbel sampling, we conduct an ablation by varying the number of samples $M$. Empirical analysis in Table 5 reveals that the performance increases sublinearly with $M$, demonstrating the expected exploration-computation trade-off. Besides, the marginal gain becomes statistically insignificant when $M \geq 10$. For comparison, we also implement GA-Gumbel, an variant of GA incorporating the Gumbel-Sinkhorn sampling. Under identical experimental protocols, GA-Gumbel demonstrates consistent performance gains over baseline GA. However, our proposed NGA still significantly outperforms GA-Gumbel, highlighting the necessity of our parameterization of the temperature.

Finally, to demonstrate the superiority of the product parameterization $\beta_l$ in NGA, we introduce two variants: (1) NGA-Manual employs a manually configured temperature schedule initialized with negative values that gradually transitions to positive values, aligned with the mean parameter distribution patterns observed in Fig. 11. (2) NGA-Scalar parameterizes temperature as a learnable scalar. In Table 5, both variants exhibit suboptimal performance, primarily due to constrained adaptive capacity and limited expressive power. This performance gap underscores the importance of our design. Note that MLP is composed of learnable weights, which is equivalent to our original implementation.

Table 5: Results of Ablation Study in terms of Accuracy (%) ↑ for solving MCES problem.

| Dataset | AIDS | MOLHIV | MCF-7 |
|---|---|---|---|
| GA | 70.99 | 71.09 | 72.92 |
| GA-Gumbel | 75.08 | 74.46 | 76.12 |
| NGA-w/o GNN | 97.45 | 98.77 | 97.11 |
| NGA-Positive | 83.42 | 85.63 | 80.32 |
| NGA-Manual | 74.27 | 70.90 | 76.45 |
| NGA-Scalar | 77.48 | 82.27 | 72.94 |
| NGA | **98.64** | **99.20** | **97.94** |

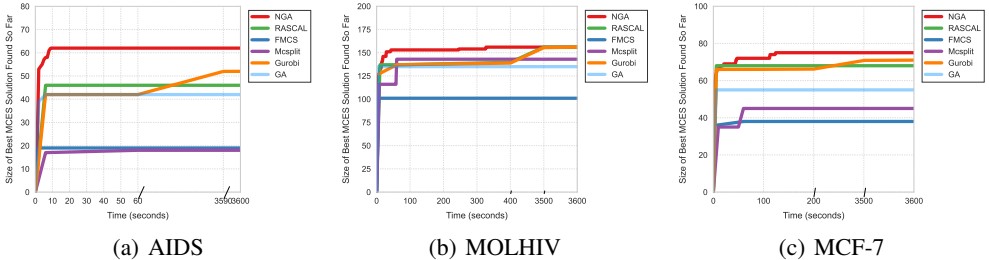

| (a) AIDS | (b) MOLHIV | (c) MCF-7 |
|---|---|---|

Figure 8: Comparison of the best MCES solution size found so far on different dataset.

## G ADDITIONAL EXPERIMENTAL RESULTS

### G.1 ADDITIONAL RESULTS FOR SOLVING MCES

We have additionally conducted a detailed analysis of the quality of the solutions obtained by our proposed NGA. In particular, we examine how frequently the MCES identified by NGA coincides with the true optimal solution. This evaluation provides deeper insights into the effectiveness and reliability of NGA beyond average performance metrics. The results highlights not only the overall accuracy of the method but also its stability across different problem instances. A summary of these results is provided in Table 6.

Table 6: The percentage of optimal solutions of MCES obtained on different datasets.

| AIDS | MCF-7 | MOLHIV |
|---|---|---|
| 49% | 56% | 61% |

### G.2 RESULTS FOR CASE STUDY

In this section, we select several instances from the three datasets as case studies. We give a time budget of 3600 seconds and record the best solution found so far for different methods, as shown in Figure 8. On the AIDS and MCF-7 datasets, our method can find better solutions within the time budget. On the MOLHIV dataset, both NGA and RASCAL find the best solution, but NGA takes much shorter time.

Specifically, we take AIDS dataset as an example and give detailed illustration about how optimization pushes the solution found so far to approximate the optimal MCES. As shown in Figure 9 , the current identified common edge subgraph serves as a lower bound, which is gradually improved as the optimization process goes.

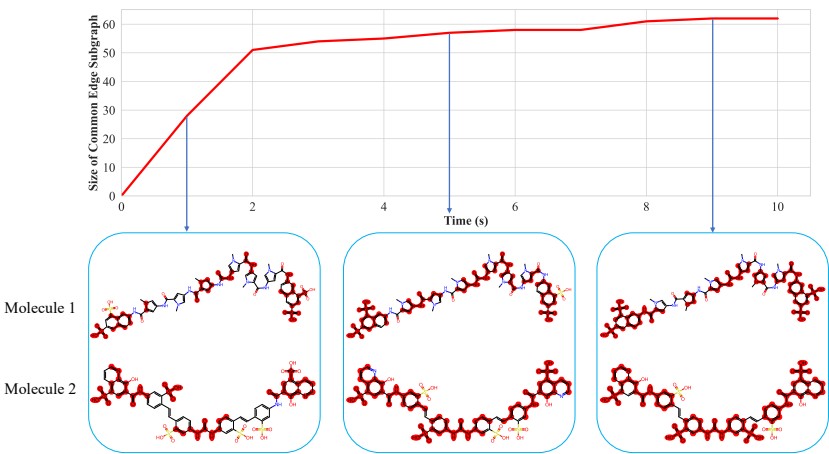

Figure 9: The size of the common edge subgraph found by NGA (serving as a lower bound) increases over the course of the optimization process (runtime in seconds). Matched atoms and bonds are highlighted in red.

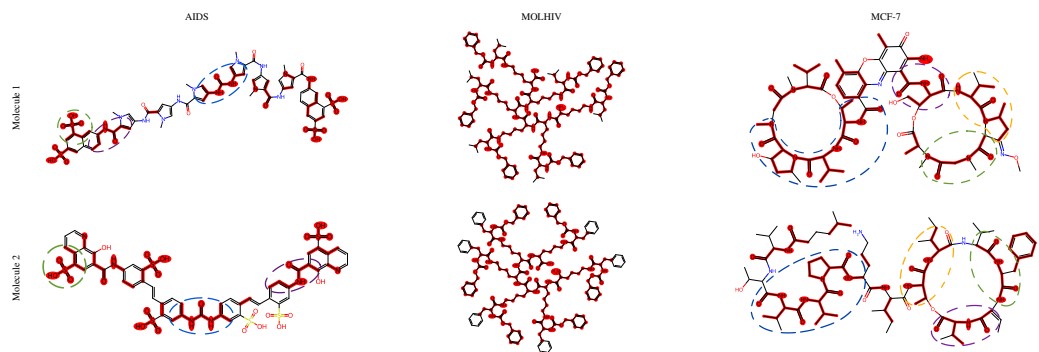

Figure 10: Visualization results of MCES for RASCAL.

### G.3 EMPIRICAL EVIDENCE

We conduct extensive experiments to demonstrate the empirical evidence of the effectiveness of our proposed method. The experiments are performed on three datasets, and all test instances are included. Specifically, we measure the parameterized temperature at different iterations during the training process, particularly after the model has converged. The learned sequence of temperatures across layers represents the optimization trajectory of the problem. This trajectory includes both positive and negative values, where the negative values often play a critical role in escaping local optima.

As shown in Figure 11, the parameterized temperature varies with different iterations in NGA, revealing stable trends across the datasets. In the initial iterations of NGA, we observe a cautious exploration of the parameter space, characterized by relatively small and stable values of the parameterized temperature. As the NGA iteration progresses, the algorithm transitions to a more aggressive convergence strategy, reflected by a significant increase in the magnitude of the parameterized temperature. Our empirical results consistently align with the theoretical analysis, highlighting distinct patterns along the optimization trajectory. These patterns reflect the dynamic behavior of the algorithm, where early stages focus on exploration and later stages emphasize convergence, guided by the learned temperature sequence.

Importantly, the values shown in Figure 11 correspond to converged or near-converged states. We additionally visualize how $|\beta_l|$ changes in the optimization process in Figure 12. Empirically, we observe that $|\beta_l|$ takes smaller magnitudes during the early and intermediate iterations—precisely when

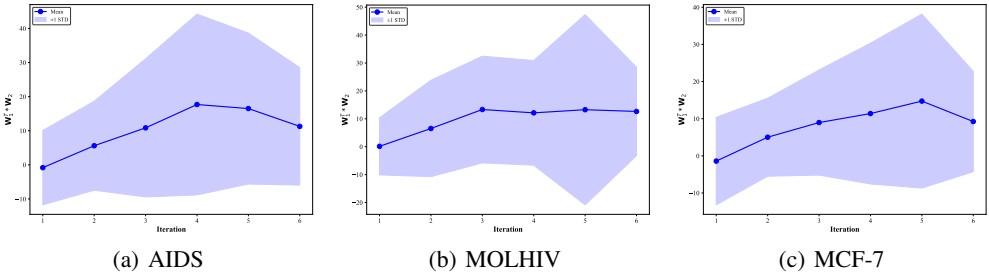

Figure 11: Distribution of parameterized temperature values across iterations in NGA on different datasets.

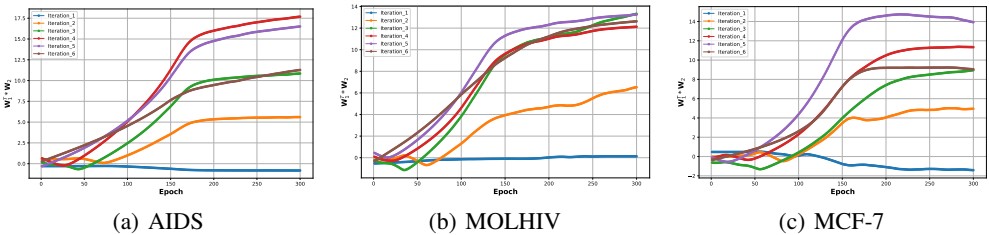

Figure 12: Mean values of parameterized temperature in different NGA iteration layers across epoch on different datasets.

escaping shallow local optima is relevant—while larger magnitudes emerge in the final refinement phase, consistent with the intended design of NGA. This aligns with the theoretical role of Theorem 1, which analyzes updates around a stationary point rather than global dynamics across all layers.

### G.4 ADDITIONAL PARAMETER ANALYSIS

To systematically evaluate the sensitivity of our method to Gumbel-Sinkhorn sampling, we conducted an ablation study by varying the number of samples $M$. Empirical analysis in Figure 14 reveals that the performance increases sublinearly with $M$, demonstrating the expected exploration-computation trade-off. Besides, the marginal gain becomes statistically insignificant when $M \geq 10$. Based on these findings, we select $M = 10$ as the optimal trade-off.

### G.5 RESULTS ON QAP INSTANCES

To demonstrate the applicability of our method to address the general QAP problem, we conducted additional experiments on Koopmans-Beckmann's QAP benchmarks: QAP20, QAP50, and QAP100,

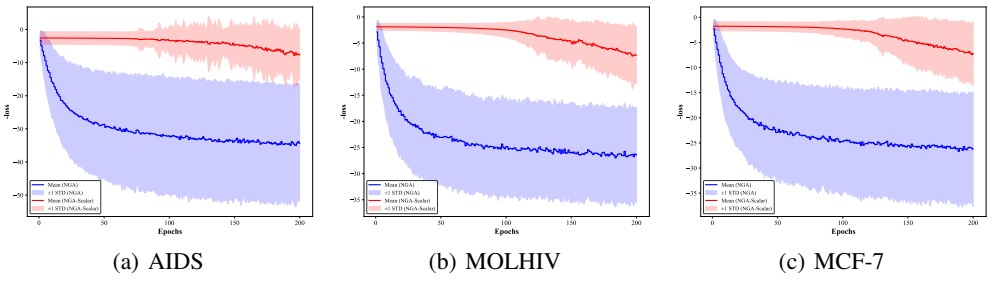

Figure 13: Loss curves across epochs on different datasets. NGA-Scalar represents an variant of NGA by setting the temperature value as a learnable scalar.

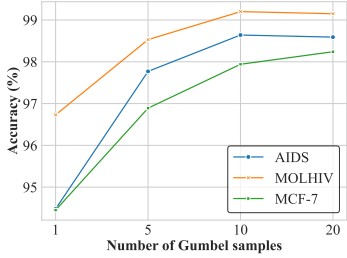

Figure 14: How the number of Gumbel samples affect NGA.

corresponding to problems with 20, 50, and 100 nodes. In each task, location node coordinates are uniformly sampled from the unit square $[0, 1]^2$. The flow $f_{ij}$ between facility $i$ and $j$ is sampled uniformly from $[0, 1]$, symmetrized, diagonal entries set to zero, and randomly set to zero with probability $p = 0.7$. We train with up to 5120 instances and evaluate on a test set of 256 instances from the same distribution. Since these tasks do not involve the MCES-specific structure, we omit ACG construction and directly use the NGA module (Alg. 1). We compare our method with the heuristic solver SM (Leordeanu & Hebert, 2005) and the learning-based solver NeuOpt (Ma et al., 2023). The results are summarized in Table 7. Our method achieves competitive or superior solutions across these QAP instances.

Table 7: Total Assignment Cost (lower is better) on Koopmans-Beckmann's QAP benchmarks

| Method | QAP20 | QAP50 | QAP100 |
|--------|-------|-------|--------|
| SM | 70.07 | 446.62 | 1800.75 |
| NeuOpt | 61.37 | 430.61 | 1789.22 |
| NGA | 62.17 | 407.59 | 1726.54 |

Besides, we conduct additional experiments on a subset of QAPLIB instances across different categories. The compared methods include GLAG (Fiori et al., 2013), PATH (Zaslavskiy et al., 2008), FAQ (Vogelstein et al., 2015). The results are summarized in Table 8. These results show that while our method is primarily designed for MCES, it can still perform competitively on general QAP, further demonstrating the versatility of our approach.

Table 8: Total Assignment Cost (lower is better) on 16 Benchmark Examples of the QAPLIB Library

| QAP | PATH | GLAG | FAQ | NGA |
|-----|------|------|-----|-----|
| chr12c | 18048 | 61430 | 13088 | 17598 |
| chr15a | 19086 | 78296 | 29018 | 17352 |
| chr15c | 16206 | 82452 | 11936 | 20978 |
| chr20b | 5560 | 13728 | 2764 | 3366 |
| chr22b | 8500 | 21970 | 8774 | 8580 |
| esc16b | 300 | 320 | 314 | 292 |
| rou12 | 256320 | 353998 | 254336 | 251094 |
| rou15 | 391270 | 521882 | 371458 | 389048 |
| rou20 | 778284 | 1019622 | 759838 | 813576 |
| tai10a | 152534 | 218604 | 157954 | 145270 |
| tai15a | 419224 | 544304 | 397376 | 418618 |
| tai17a | 530978 | 708754 | 520754 | 537896 |
| tai20a | 753712 | 1015832 | 736140 | 802464 |
| tai30a | 1903872 | 2329604 | 1908814 | 1861206 |
| tai35a | 2555110 | 3083180 | 2531558 | 2789834 |
| tai40a | 3281830 | 4001224 | 3237014 | 3214724 |

Table 9: MCES results on unlabeled dataset

| Dataset | AIDS-unlabeled |
|---------|----------------|
| RASCAL | 89.74 |
| FMCS | 61.53 |
| Mcsplit | 74.47 |
| GLSearch | 68.13 |
| NGM | 55.20 |
| GA | 67.44 |
| Gurobi | 83.89 |
| NGA | 93.72 |
| NGA-GPS | 94.37 |

### G.6 RESULTS ON UNLABELED DATASETS

To further enhance the generalizability of our method, we have additionally conducted experiments on unlabeled graphs. Specifically, we constructed an unlabeled version of the AIDS dataset, denoted as AIDS-unlabeled, where all atom types were replaced by carbon atoms (C) and all bonds were converted to single C–C bonds. This transformation results in unlabeled graphs while preserving the overall structural diversity of the original dataset.

In the unlabeled settings, additional structural embeddings may provide meaningful information. We replaced the original GCN backbone in our method NGA with a Graph Transformer architecture equipped with structural encodings (Rampášek et al., 2022) and name this variant as NGA-GPS. This variant naturally incorporates positional and structural information, enabling NGA to operate effectively when explicit labels are absent.

The experimental settings follow those used in our paper (Appendix 6.2). For NGA and NGA-GPS, we employ an 8-layer GNN with 32 hidden dimensions. The NGA-GPS is equipped with random walk based structural encodings and the multi-head attention mechanism with 4 heads. For the other baseline methods, we also adhere to the configurations described in our paper. The experimental results in Table 9 show that our method maintains strong performance even in the unlabeled setting, demonstrating its robustness and potential applicability beyond labeled molecular graphs.

## H  LIMITATIONS AND FUTURE WORK

This work focuses on labeled graphs, leaving open the question of how to handle unlabeled ones. When node or edge labels are absent, learning correspondences relies solely on graph structure, which increases the number of potential matches per node, enlarges the solution space, and makes the problem significantly more challenging.

Multi-graph MCES is a meaningful and practically relevant extension of the classical pairwise MCES formulation. In this work, we focus on the pairwise setting, which is the standard and well-established definition of MCES and also a necessary first step toward addressing the long-standing and more general challenge. Pairwise MCES provides the fundamental building block on top of which multi-graph formulations are typically constructed. A feasible path toward multi-graph MCES is to combine pairwise MCES computations with cycle-consistency constraints, which has been long studied in multi-graph matching (Pachauri et al., 2013; Xia et al., 2025). Our method can be extended along these lines—by running NGA across all graph pairs and enforcing consistency—to obtain a principled multi-graph solution. We view this direction as a natural and promising extension of our work.

## I  DISCUSSION ON GRAPH LABELS

In the MCES formulation, a common subgraph requires exact matching of compatible labels. Therefore, the choice of labels directly reflects what we consider as "common structure." Labels can be domain-specific features aligned with the MCES problem definition.

Concrete examples across domains:

- Molecular graphs: Atom types (C, N, O) and bond types (single, double, aromatic) are natural labels because pharmacological similarity requires exact chemical structure matching.
- Social networks: User attributes (occupation, location, age group) could serve as labels when finding common community structures.
- Knowledge graphs: Entity types and relation types naturally define what constitutes matching substructures.

If the application requires tolerance to label noise or similarity rather than exact matching, one could:

- Preprocess labels by clustering fine-grained categories into coarser groups
- Define softer compatibility functions in the ACG construction (modifying Eq. 4 to allow approximate matching)
- Formulate as a different problem variant (e.g., approximate common subgraph)

## J   LLM USAGE STATEMENT

During the preparation of this manuscript, we utilized a large language model (LLM) as a writing assistant. The LLM's role was strictly limited to improving the clarity, grammar, and readability of our text through sentence polishing and paragraph restructuring. The LLM did not contribute to research ideation, experimental design, data analysis, or the formulation of conclusions. All scientific content and claims are the sole responsibility of the human authors.

