# OpenReview forum: "Neural Graduated Assignment for Maximum Common Edge Subgraphs"
_ICLR.cc/2026/Conference — ICLR 2026 Poster_

### Official Review · Reviewer_xBg1 · 2025-10-27

**Soundness:** 3
**Presentation:** 3
**Contribution:** 3
**Rating:** 6
**Confidence:** 3

**Summary:**

This paper introduces the *Neural Graduated Assignment* (NGA), a neural optimization framework designed to solve the MCES problem efficiently. Traditional algorithms for solving MCES suffer from scalability issues, especially when dealing with large graphs. The NGA approach overcomes these limitations by leveraging a learnable temperature mechanism and unsupervised training, allowing the algorithm to scale efficiently and avoid computational bottlenecks. The paper provides theoretical justification for the efficacy of NGA, demonstrating its ability to balance exploration and exploitation in the search process. Experimental results show that NGA significantly improves computation time, scalability, and performance compared to existing methods, making it a promising solution for MCES and related problems.

**Strengths:**

1. Novel Approach: NGA represents advancement in solving the MCES problem by integrating neural components with traditional optimization frameworks. The use of a learnable temperature schedule and the end-to-end trainability of the model is an innovative approach.

2. Theoretical Foundation: The authors provide a strong theoretical analysis of NGA, including its convergence properties and the mechanisms through which it escapes local optima. This adds robustness to the method and makes it more reliable in practical use.

3. Extensive Experiments: The paper includes thorough experimentation across multiple datasets and tasks, such as MCES computation, graph similarity estimation, and graph retrieval, showing that NGA outperforms existing methods in terms of both accuracy and computational efficiency.

**Weaknesses:**

1. The paper's reliance on modeling MCES as a QAP leads to O(N²) space complexity due to the affinity matrix. For large graphs (e.g., those with more than 100 nodes), this could pose significant memory challenges, making the method less feasible for very large-scale graphs. The paper does not provide an analysis of memory usage, which is a crucial aspect when considering practical applications for large datasets.

2. While the paper claims that NGA is interpretable, the exact mechanisms through which the model learns to assign graph correspondences are not fully clear. The interpretability of the learned assignments, especially in complex graph structures, could be more thoroughly discussed.

3. While the paper compares well against traditional solvers and some learning-based graph matching models (NGM), it could be strengthened by a comparison to other recent neural approaches for combinatorial optimization on graphs, e.g., unsupervised methods for graph matching.

**Questions:**

1. Could you provide more details on how the learnable temperature schedule in NGA compares to other dynamic temperature annealing methods in optimization? Is there a specific advantage in using this formulation over others?

2. Could you provide an analysis of CPU/GPU memory usage w.r.t. the scale of graphs?

3. How does the performance of NGA change when applied to graphs with significant amounts of noise or missing data? How would NGA perform in scenarios with highly asymmetric graph pairs (e.g., when the graphs have very different numbers of nodes/edges)? Could you analyze the robustness of NGA?

4. What impact does the choice of neural network architecture have on the overall performance of NGA? Would other architectures, like graph transformers, yield better results?

---

> ### Author Response · Authors · 2025-11-18
>
> Thank you for your valuable comments, as well as for acknowledging our contributions. Below, we respond to your specific comments.
>
> > W1 & Q2
>
> The overall space complexity of modeling MCES as a QAP leads to $O(|\mathcal{V}_1|\times |\mathcal{V}_2|+|\mathcal{E}_1|\times |\mathcal{E}_2|)$ space complexity.  Empirically, we find this process to be highly memory efficient. For example, when selecting two molecular graphs (214 and 163 nodes) from the AIDS dataset, the tensors related to the QAP formulation take 22 MB memory.
>
> Since our method primarily runs on GPU, we further analyze how the total GPU memory consumption (including the model parameters) scales with graph size during training. For graph pairs with approximately 30 nodes, the memory usage is 429 MB. For graph pairs with around 100 nodes, the usage increases to 1209 MB, and for those with roughly 200 nodes, the usage is 2405 MB.
>
> These results confirm that our approach is highly memory efficient and imposes very modest hardware requirements.
>
> > W2
>
> We appreciate the reviewer’s thoughtful suggestion regarding interpretability. In this work, the interpretability claim refers specifically to the fact that, through the construction of the ACG, the learned assignments produced by NGA can be directly mapped to __common subgraphs__, providing a clear structural meaning to the model’s outputs. A more detailed explanation of this mechanism is provided in Appendix B, with an schematic illustration in Fig. 6.
>
> We thank the reviewer for pointing out that additional clarification would be helpful. In response, we have expanded the discussion in Section 4.2.1 and Appendix B to more thoroughly illustrate how the ACG exposes the interpretable structure underlying the learned correspondences.
>
> > W3
>
> We thank the reviewer for this helpful suggestion. In addition to the baselines reported in the main paper, we have included a comparison with GANN-GM [1], a recent unsupervised graph matching method. The corresponding results are now provided below.
>
> |  | AIDS   | MOLHIV | MCF-7  |
> |:----------|:-------:|:-------:|:-------:|
> | GANN-GM   | 49.76  | 72.18  | 63.47  |
>
> As shown in the experiments, the performance of GANN-GM on MCES is relatively limited. One possible reason is that its unsupervised learning signal relies on a traditional graph matching solver to provide supervisory guidance. Since these classical solvers themselves struggle on the MCES task, the resulting signal may be too weak or noisy to support effective learning.
>
> We hope this additional comparison helps clarify how NGA differs from existing unsupervised approaches and highlights the challenges of applying solver-based self-supervision to MCES.
>
> [1] Wang, Runzhong, Junchi Yan, and Xiaokang Yang. "Unsupervised learning of graph matching with mixture of modes via discrepancy minimization." IEEE Transactions on Pattern Analysis and Machine Intelligence 45.8 (2023): 10500-10518.
>
> > Q1
>
> We appreciate the reviewer’s interest in how our learnable temperature schedule compares with other dynamic annealing strategies. To provide a clearer picture, we include comparisons in Appendix F with both (i) a learnable scalar temperature parameterization (NGA-Scalar) and (ii) a non-learnable but dynamic annealing scheme (GA).
>
> Compared with these alternatives, our formulation offers two key advantages. First, as shown in Proposition 2, the product parameterization allows the model to __adaptively adjust the effective learning rate during optimization__. Second, it naturally induces an __accelerating effect__ on the magnitude of updates to the temperature parameter, which we find beneficial for escaping shallow local minima while still enabling stable convergence. These behaviors are confirmed empirically in Appendix D.2.
>
> We hope this additional clarification helps articulate the motivation and benefits of the proposed temperature schedule.
>
> > Q3
>
> We appreciate the reviewer’s thoughtful questions regarding robustness. The effects of noise and missing data are discussed in our response to __Reviewer 6SDK Q3__, where we provide additional analysis on how NGA behaves under such perturbations.
>
> Regarding asymmetric graph pairs, we would like to note that in our experimental setup, each graph pair is selected __randomly__ from the dataset. As a consequence, many instances naturally involve substantial asymmetry. In fact, based on our statistics, approximately 25% of the graph pairs differ in the number of nodes by a factor of 1.5× or more. Across these cases, NGA does not exhibit noticeable sensitivity, and its performance remains stable in the overall results.
>
> We hope this clarification helps provide a better picture of the robustness properties of NGA.

---

> ### Author Response · Authors · 2025-11-18
>
> > Q4
>
> We appreciate the reviewer’s question regarding the choice of neural network backbone. During method development, we experimented with several architectures, including deeper GNNs and more expressive variants. However, the performance differences were generally quite small, which led us to adopt the simplest GCN backbone in the final version. One possible reason is that molecular graphs contain rich node and edge features, so the marginal benefit of using more complex architectures may be limited in this setting.
>
> That said, as discussed in our response to __Reviewer iuLj W1 & Q2__, in the unlabeled graph scenario, we do observe that using architectures such as Graph Transformers can yield noticeable performance gains. This suggests that the optimal choice of backbone may depend on the specific characteristics of the input graphs.
>
> We hope this additional clarification provides more context on how architectural choices affect NGA’s performance.

---

> > ### Comment · Reviewer_xBg1 · 2025-11-19
> >
> > Thank you for your rebuttal. I have no further questions.

---

> > > ### Author Response · Authors · 2025-11-19
> > >
> > > Thank you for taking the time to carefully review our additional responses. We sincerely appreciate your continued positive assessment and your support for our work. Your thoughtful feedback has been invaluable in helping us refine our paper, and we are grateful for your recommendation.

---

### Official Review · Reviewer_6SDK · 2025-10-27

**Soundness:** 4
**Presentation:** 3
**Contribution:** 3
**Rating:** 8
**Confidence:** 2

**Summary:**

This work tackles the Maximum Common Edge Subgraph (MCES) problem: given two labeled graphs, the goal is to find the largest shared subgraph with the maximum number of common edges. This problem is especially relevant in molecular analysis and is NP-complete. The proposed method, Neural Graduated Assignment (NGA), is an unsupervised, learnable approximation with three main components: (i) it constructs an Association Common Graph (ACG) that only considers node and edge pairs that are label-compatible; (ii) it learns a soft node-to-node assignment that is iteratively refined using a learnable, high-dimensional temperature, which the authors argue improves convergence behavior; and (iii) it discretizes this soft assignment into a final match, using multiple sampled candidates at inference time to encourage exploration. The experiments evaluate NGA both on MCES quality and on downstream tasks such as graph similarity prediction and graph retrieval. NGA outperforms prior baselines under time-constrained scenarios on molecular datasets. Ablation studies support several design choices (use of high-dimensional temperature, sampling strategy, etc.) and provide insight into optimization dynamics.

**Strengths:**

- The method explicitly balances exploration and exploitation of the solution space.
- The paper provides a clear analysis of the optimization dynamics, which gives provable guarantees for convergence.
- The experiments are thorough: multiple tasks (MCES, similarity, retrieval), strong ablations, and consistent gains over competitive baselines.
- It provides a better trade-off between scalability and performance than the prior works.

**Weaknesses:**

1. The method is not reusable for more than one pair of graphs at a time.
2. The paper assumes molecular settings, which have particular properties: the ACG is sparse, the node and edge labels are known and meaningful, etc.
3. Runtime is mainly evaluated under a fixed time cap; it would be useful to see quality vs. time curves or difficulty-stratified results.
4. Minor: some citation formatting issues in the appendix (e.g. lines 719,855,885).

**Questions:**

1. Why would supervised methods perform badly with multiple ground truths? Don't most tasks in deep learning have multiple equally good solutions anyway?
2. How do the methods compare in number of learnable parameters?
3. Can the labels/features be anything? How would the method perform if they are e.g. too fine-grained or noisy? For instance, in molecular datasets we know that the atom types are a good label, but in other instances we might not know what to choose.
4. Relatedly, if the graphs are unlabeled, would it make sense to create the labels by another procedure (e.g. by structural cues) and then run the method? Would this perform better than not giving them any labels?
5. How does complexity change if the ACG is not sparse? In which (non-molecular) settings can that occur?
6. Which (non-molecular) domains are other realistic targets for NGA, and which remain out of scope under the current assumptions?

---

> ### Author Response · Authors · 2025-11-18
>
> Thank you for your insightful feedback. We are happy to address your concerns and questions. Detailed responses to your comments are provided below.
>
> > W1
>
> We sincerely thank the reviewer for raising this important point. Indeed, multi-graph MCES is a meaningful and practically relevant extension of the classical pairwise MCES formulation.
>
> In this work, we focus on the pairwise setting, which is the standard and well-established definition of MCES and also a necessary first step toward addressing this long-standing and more general challenge. Pairwise MCES provides the fundamental building block on top of which multi-graph formulations are typically constructed.
>
> A feasible path toward multi-graph MCES is to combine pairwise MCES computations with __cycle-consistency constraints__, which has been long studied in multi-graph matching [1-2]. Our method can be extended along these lines—by running NGA across all graph pairs and enforcing consistency—to obtain a principled multi-graph solution. We view this direction as a natural and promising extension of our work.
>
> We appreciate the reviewer for highlighting this opportunity, and we plan to explore multi-graph MCES more thoroughly in future work.
>
> [1] Pachauri, Deepti, Risi Kondor, and Vikas Singh. "Solving the multi-way matching problem by permutation synchronization." Advances in neural information processing systems 26 (2013).
>
> [2] Xia, Yifan, et al. "Multi-Shape Matching with Cycle Consistency Basis via Functional Maps." Proceedings of the AAAI Conference on Artificial Intelligence. Vol. 39. No. 8. 2025.
>
> > W2 & Q4
>
> We appreciate the reviewer’s thoughtful observation. MCES has indeed been historically defined and widely studied in the molecular domain, where maximum common substructure search is a central problem. For this reason, evaluating our method on molecular graphs is a natural starting point that aligns with the original formulation of MCES.
>
> Regarding the reviewer’s comment on sparsity and meaningful labels, we would like to gently note that these characteristics are not exclusive to molecular graphs. Many real-world graphs are similarly sparse, and a variety of domains incorporate informative node or edge features. Nonetheless, we understand the reviewer’s concern about generality and have therefore included additional experiments on __unlabeled graphs__ (see our response to __Reviewer iuLj W1 & Q2__ and our update of manuscript in Appendix G.6).
>
> On the question of creating synthetic labels from structural cues, our observations suggest that introducing artificial labels may unintentionally restrict the problem, as such labels could encode assumptions that are not necessarily consistent with the underlying MCES objective. Instead, NGA combined with positional or structural information functions effectively in the absence of explicit labels, providing a more flexible and task-aligned solution.
>
> We hope this clarification addresses the reviewer’s questions, and we are grateful for the opportunity to elaborate on this point.
>
> > W3
>
> We appreciate the reviewer’s suggestion. The quality–vs.–time curves are already provided in Appendix G.2, specifically in Fig. 8, where we report how solution quality evolves over time across different settings. We also include several visualizations of solutions across time in Fig. 9. These results complement the fixed–time–cap evaluation and offer a more fine-grained view of how NGA behaves over time. We will highlight this more clearly in the revised version.
>
> > W4
>
> We thank the reviewer for pointing out the citation formatting issues in the appendix. We have corrected them in the revised version.
>
> > Q1
>
> Supervised graph matching methods rely on one fixed target assignment for training. However, in the case of MCES, obtaining reliable ground-truth matchings is itself extremely challenging, as different maximum common subgraphs can be structurally distinct yet equally valid. As a result, the supervision becomes inherently inconsistent: the model is penalized for predicting any correct solution that does not coincide with the arbitrarily chosen label. This leads to conflicting gradients and degraded performance.
>
> This situation is fundamentally different from typical deep learning tasks where multiple good solutions exist at the parameter level: in standard supervised learning, the labels themselves are unique and consistent, even though many parameter configurations may achieve the same loss. In contrast, for MCES, the labels (target matchings) are not unique, and supervised training forces the model to fit an underdetermined and inconsistent target space.
>
> > Q2
>
> A detailed comparison of learnable parameters is provided in Section 6.4, with results summarized in Fig. 4(b). As shown there, when the hidden dimension $d$ is small, the capacity of the model is limited. As $d$ increases, the number of learnable parameters grows accordingly, and the performance steadily improves until it saturates at higher dimensions.

---

> ### Author Response · Authors · 2025-11-18
>
> > Q3
>
> Thank you for this important question about the flexibility and robustness of our method with respect to label choices. Labels can be domain-specific features aligned with the MCES problem definition. The key principle is that labels should encode the semantic properties that define "compatibility" between nodes/edges in the two graphs. In the MCES formulation, a common subgraph requires exact matching of compatible labels. Therefore, the choice of labels directly reflects what we consider as "common structure."
>
> __Concrete examples across domains:__
> - Molecular graphs: Atom types (C, N, O) and bond types (single, double, aromatic) are natural labels because pharmacological similarity requires exact chemical structure matching.
> - Social networks: User attributes (occupation, location, age group) could serve as labels when finding common community structures.
> - Knowledge graphs: Entity types and relation types naturally define what constitutes matching substructures.
>
> __Regarding fine-grained or noisy labels:__ If labels are too fine-grained or contain noise such that corresponding nodes have subtle differences, then __by the MCES problem definition itself__, these nodes should __NOT__ be considered as part of the common subgraph. This is not a limitation of our method, but rather an inherent characteristic of the MCES problem. For instance, if noise causes label mismatch, the structures are fundamentally different under the given labeling scheme and thus become a new pair of MCES problem instance.
>
> __This affects all methods uniformly.__ Both our NGA and baseline search algorithms (RASCAL, McSplit, etc.) operate under the same MCES definition—they all require exact label compatibility. The performance degradation under noisy labels would be consistent across methods, as it stems from the problem formulation rather than the algorithmic approach.
>
> If the application requires tolerance to label noise or similarity rather than exact matching, one could:
> 1. Preprocess labels by clustering fine-grained categories into coarser groups
> 2. Define softer compatibility functions in the ACG construction (modifying Eq. 4 to allow approximate matching)
> 3. Formulate as a different problem variant (e.g., approximate common subgraph)
>
> We appreciate this comment and have added a discussion on label selection guidelines in Appendix I in the revised manuscript.
>
> > Q5
>
> The construction of the ACG requires enumerating all edge pairs between the two input graphs $G_1$ and $G_2$. Assume that the constructed ACG has $M$ edges. In the worst case where all edge labels are compatible (or graphs are unlabeled), the ACG can contain up to $M = |\mathcal{E}_1|\times |\mathcal{E}_2|$ edges. leading to a time complexity of $O(|\mathcal{E}_1|\times |\mathcal{E}_2|)$.  If the graphs are not sparse, e.g. for graphs with average degree $k$, we have $|\mathcal{E}_1| \approx k\cdot |\mathcal{V}_1| / 2$ and $|\mathcal{E}_2| \approx k\cdot |\mathcal{V}_2| / 2$, yielding $O(|\mathcal{V}_1| \cdot |\mathcal{V}_2| \cdot k^2)$ complexity. The worst case happens when two graphs are both densely connected and all nodes and edges have compatible labels.
>
> > Q6
>
> Beyond molecular datasets, NGA is well-suited to domains where node labels encode meaningful structural semantics. Realistic non-molecular targets include task assignment issues where source and target graphs are task interaction graph and processors graph [1], and protein–interaction networks where residue or domain annotations act as labels [2]. In such domains, discrete and semantically aligned labels allow NGA to reliably match substructures.
>
> Under the current assumptions, NGA is less suitable for graphs that are extremely dense or highly dynamic. These settings violate the requirement for meaningful graph structures that support stable matching.
>
> [1] Bahiense, Laura, et al. "The maximum common edge subgraph problem: A polyhedral investigation." Discrete Applied Mathematics 160.18 (2012): 2523-2541.
>
> [2] Ehrlich, Hans‐Christian, and Matthias Rarey. "Maximum common subgraph isomorphism algorithms and their applications in molecular science: a review." Wiley Interdisciplinary Reviews: Computational Molecular Science 1.1 (2011): 68-79.

---

> ### Comment · Reviewer_6SDK · 2025-11-24
>
> W1/W2: Thank you for adding extra information about these topics.
>
> W3: I apologize for not seeing the plots, I am happy with it as is.
>
> Q2: Apologies for the confusion, I didn't mean parameter growth compared to itself (increase in d), but to the other architectures that NGA is evaluated against. I was curious to see where they lie.
>
> Q3: Thank you for adding the extra discussion, I find this important. I like the relationship to knowledge graphs and information recovery. I see it as a possible future direction too, as I am sure there are particular challenges to this setting.
>
> Extra Q: I thank reviewer iuLj for bringing up to my attention the product parameterization. Previous work in the implicit bias literature has studied this problem of quadratic reparameterizations explaining the dynamic learning rate of $\beta$ based on initalization, and I recommend that the link is established in the revised manuscript, as it would give more depth to this choice. Some foundational papers:
>
> (1) Li, Zhiyuan et al. “Implicit Bias of Gradient Descent on Reparametrized Models: On Equivalence to Mirror Descent.” NeurIPS (2022).
>
> (2) Jacobs, Tom et al. “Mirror, Mirror of the Flow: How Does Regularization Shape Implicit Bias?” ICML (2025).
>
> And its application to sparse training, which has optimization challenges as it needs the smooth approximation of the LASSO objective:
>
> (3) Jacobs, Tom and Rebekka Burkholz. “Mask in the Mirror: Implicit Sparsification.” ICLR (2025).
>
> (4) Kolb, Chris et al. “Deep Weight Factorization: Sparse Learning Through the Lens of Artificial Symmetries.” ICLR (2025).

---

> > ### Author Response · Authors · 2025-11-26
> >
> > Thank you for taking the time to review our response and for your thoughtful feedback throughout this process. We are glad to hear that our explanation clarified our approach. Detailed responses to your comments are provided below.
> >
> > > Q2
> >
> > We thank the reviewer for the clarification. We apologize for our earlier misunderstanding. For completeness, we have now added a summary of the parameter sizes of learning-based baseline architectures evaluated in our experiments. For the baseline methods NGM and GANN-GM, we follow their original configurations and use 16 feature channels by default. We additionally increase the number of feature channels to 32 and 64, and report the results below. As shown in the table, neither method exhibits noticeable performance improvement as the model size increases.
> >
> > |                  |    |   AIDS    |       | |  |   MOLHIV    |       ||   |    MCF-7   |       |
> > |------------------|:------:|:-----:|:-----:|:-----:|:------:|:-----:|:-----:|:-----:|:------:|:-----:|:-----:|
> > | Feature channels |   16   |   32  |   64  ||   16   |   32  |   64  ||   16   |   32  |   64  |
> > | NGM              |  33.34 | 33.21 | 33.86 ||  52.21 | 52.34 | 53.07 ||  42.38 | 42.76 | 42.59 |
> > | GANN-GM          |  49.76 | 49.78 | 49.43 ||  72.18 | 72.32 | 72.09 ||  63.47 | 63.86 | 63.77 |
> >
> > > Extra Q
> >
> > We sincerely thank the reviewer for pointing out these highly relevant works on  reparameterized gradient dynamics, as well as for highlighting the connection to quadratic/product parameterizations. These references are indeed valuable, and we appreciate the suggestion to relate our formulation to the broader implicit-bias perspective.
> >
> > In the revised manuscript (Appendix D.2), we have included a discussion acknowledging this line of work [1-2] and how it provides additional conceptual depth to our product parameterization. We also appreciate the reviewer’s pointers to the recent developments on sparsification and deep weight factorization [3-4], which offer further context on the role of reparameterizations in shaping optimization dynamics.
> >
> > While a full theoretical integration is beyond the scope of the present paper, we fully agree that drawing this connection enriches the motivation behind our choice of parameterization, and we will incorporate a brief discussion to acknowledge these ties in a balanced and accurate way.
> >
> > We are grateful to the reviewer for bringing these insightful references to our attention.

---

> > > ### Comment · Reviewer_6SDK · 2025-11-26
> > >
> > > Thank you for the readiness in providing tangible answers to my questions. I am happy with my original positive assessment of the work so I will maintain my score, but I will increase my confidence. RE: Extra Q, some of the new citations do not have the year, which looks slightly strange in-text.

---

> > > > ### Author Response · Authors · 2025-11-27
> > > >
> > > > We sincerely thank you for your positive assessment and for increasing your confidence score. We truly appreciate your supportive feedback and the helpful suggestions throughout the review process. Your comments provide valuable insights that have helped us improve the paper in many ways.
> > > >
> > > > Regarding the extra citations, we apologize for the missing publication years in the in-text references. We have corrected these formatting issues in the revised manuscript.

---

### Official Review · Reviewer_iuLj · 2025-11-09

**Soundness:** 3
**Presentation:** 3
**Contribution:** 3
**Rating:** 6
**Confidence:** 5

**Summary:**

This paper introduces Neural Graduated Assignment (NGA), a novel unsupervised neural optimization framework for solving the Maximum Common Edge Subgraph (MCES) problem. The authors formulate MCES as a Quadratic Assignment Problem (QAP) via the construction of an Association Common Graph (ACG). The core innovation lies in replacing the fixed temperature parameter in classical Graduated Assignment (GA) with a learnable, high-dimensional temperature parameterization, which enables adaptive exploration and exploitation during optimization. The method is unsupervised, scalable, and theoretically analyzed for its convergence and local optima escape behavior. Extensive experiments on molecular datasets demonstrate that NGA significantly outperforms existing methods in accuracy, scalability, and efficiency, and shows strong performance in downstream tasks like graph similarity computation and retrieval.

**Strengths:**

1, Novel Formulation and Guarantees: The introduction of the Association Common Graph (ACG) is a crucial contribution. It provides an elegant way to ensure that any valid subgraph extracted from it is a correct common subgraph of the input graphs. This formulation cleanly transforms the MCES problem into a QAP with inherent structural guarantees, a foundational step that enables the subsequent neural optimization.
2, Rigorous Theoretical Underpinning: The paper goes beyond empirical results by providing a solid theoretical analysis. It explains how NGA escapes local optima (Theorem 1) by leveraging the variance of the gradient, and why the product parameterization accelerates convergence (Proposition 2) compared to a scalar one. This theoretical grounding significantly strengthens the methodological claims.

3, Differentiable and Adaptive Optimization Core: The proposed Neural Graduated Assignment (NGA) mechanism is the paper's central innovation. By replacing the static temperature in classical GA with a learnable, high-dimensional parameterization (  β_l = W_1^T  W_2), the method dynamically balances exploration and exploitation. This design eliminates cumbersome manual scheduling and allows the model to adapt its optimization trajectory to the specific problem instance, leading to faster convergence and better performance.

**Weaknesses:**

1.Limited Discussion on Unlabeled Graphs: The method assumes labeled graphs (node/edge features). While this is reasonable for molecular data, many real-world graphs are unlabeled or partially labeled. The paper does not discuss how NGA might be adapted to such settings, which limits its generalizability.

2, Computational Overhead of ACG: The construction of the ACG is central to the method but may become computationally expensive for very large graphs. The paper does not analyze the scalability of ACG construction in depth, nor does it compare its overhead relative to the overall optimization.

3, Theoretical Assumptions: The theoretical analysis relies on small ∣ β_l ​ ∣ assumptions (Lemma 1), which may not always hold in practice. The empirical distribution of  ∣ β_l ​ ∣  (Fig. 11) shows both small and large values, so the applicability of the theory across all layers is not fully justified.

**Questions:**

1，ACG Scalability: What is the time and space complexity of constructing the ACG? How does it scale with graph size and label dimensionality? Could approximate or sparse ACG constructions be used for very large graphs?

2， Generalization to Unlabeled Graphs: Have you considered or experimented with unlabeled graphs? Could structural embeddings (e.g., positional encodings) replace or complement label information in such cases?

3， Training Stability: The product parameterization β_l = W_1^T  W_2 can lead to unstable gradients. Did you observe such issues during training? Were any techniques (e.g., gradient clipping, normalization) used to stabilize training?

4, Choice of Parameterization: Why was the product form W_1^T  W_2 ​ chosen over other parameterizations (e.g., MLP or direct scalar)? Was this motivated by empirical performance or theoretical insights?

---

> ### Author Response · Authors · 2025-11-18
>
> We would like to express our sincere gratitude for your thorough review of our manuscript and for providing valuable feedback and suggestions. We aim to clarify important points about the problem and outline how our approach resolves the issues you raised.
>
> > W1 & Q2
>
> We appreciate the reviewer’s insightful comment. We agree that our original focus on labeled graphs is mainly motivated by the origin of the MCES problem, which arises from molecular graph analysis where node and edge labels naturally exist.
> To further enhance the generalizability of our method, we have additionally conducted experiments on unlabeled graphs. Specifically, we constructed an unlabeled version of the AIDS dataset, denoted as AIDS-unlabeled, where all atom types were replaced by carbon atoms (C) and all bonds were converted to single C–C bonds. This transformation results in unlabeled graphs while preserving the overall structural diversity of the original dataset.
>
> Besides, we agree that structural embeddings can provide meaningful information in unlabeled settings. We replaced the original GCN backbone in our method NGA with a Graph Transformer architecture equipped with structural encodings (GPS [1]) and name this variant as NGA-GPS. This variant naturally incorporates positional and structural information, enabling NGA to operate effectively when explicit labels are absent.
>
> The experimental settings follow those used in our paper (Appendix C.2). For NGA and NGA-GPS, we employ an 8-layer GNN with 32 hidden dimensions. The NGA-GPS is equipped with random walk based structural encodings and the multi-head attention mechanism with 4 heads. For the other baseline methods, we also adhere to the configurations described in our paper. The experimental results below show that our method maintains strong performance even in the unlabeled setting, demonstrating its robustness and potential applicability beyond labeled molecular graphs.
>
>
> |Method|Accuracy (%)|
> |:----------|------:|
> | RASCAL    | 89.74 |
> | FMCS      | 61.53 |
> | Mcsplit   | 74.47 |
> | GLSearch  | 68.13 |
> | NGM       | 55.20 |
> | GA        | 67.44 |
> | Gurobi    | 83.89 |
> | NGA       | 93.72 |
> | NGA-GPS   | 94.37 |
>
> We appreciate the reviewer’s suggestion, which indeed helped us strengthen the evaluation and better demonstrate the generalizability of our method. These additional results have been updated in Appendix G.6.
>
> > W2 & Q1
>
> We thank the reviewer for this thoughtful comment. As discussed in Section 4.2.1, the construction of the ACG requires enumerating all edge pairs between the two input graphs $G_1$ and $G_2$, leading to a time complexity of $O(|\mathcal{E}_1|\times |\mathcal{E}_2|)$. In practice, however, this cost is very limited because molecular graphs are typically sparse — each atom usually has only a few neighbors (typically up to 4) due to chemical valency constraints and thus $|\mathcal{E}_1|\times |\mathcal{E}_2| \ll |\mathcal{V}_1|^2 \times |\mathcal{V}_2|^2$.
>
> The space complexity of constructing the ACG is $O(M)$, where $M$ is the number of edges in ACG.  An edge in the ACG exists only when the corresponding pair of edges in  $G_1$ and $G_2$ satisfy the compatibility constraints. Consequently, the actual number of ACG edges satisfies $M < |\mathcal{E}_1|\times |\mathcal{E}_2| \ll |\mathcal{V}_1|^2 \times |\mathcal{V}_2|^2$. To handle this efficiently, we can store the ACG using sparse data structures, such as sparse tensors, which allow memory usage to scale with $O(M)$ rather than with the dense upper bound. This ensures that the space overhead remains manageable even for relatively large graphs.
>
> Empirically, we find the ACG construction to be highly efficient. For example, when selecting two molecular graphs (214 and 163 nodes) from the AIDS dataset, the ACG has 14389 edges and can be constructed with 0.163 second on our hardware setup. This indicates that the overhead of ACG construction is negligible compared with the overall optimization process.
> The analysis of time and space complexity has been updated in Appendix C.3.

---

> ### Author Response · Authors · 2025-11-18
>
> > W3
>
> We appreciate the reviewer’s observation regarding the small-$|\beta_l|$ assumption in Lemma 1. We would like to clarify that this assumption is local and is used only for analyzing the behavior of NGA near a stationary point. The theory does not require $|\beta_l|$  to remain small throughout the entire optimization trajectory.
>
> Importantly, the values shown in Fig. 11 correspond to __converged or near-converged states__. At this stage, the assignment matrix has already become nearly discrete, and $|\beta_l|$ naturally grows larger to sharpen the assignment and accelerate final convergence. Thus, the static distribution in Fig. 11 reflects only the late exploitation phase and should not be interpreted as the behavior of $|\beta_l|$ during the earlier optimization stages.
>
> To address this concern, we additionally visualize how $|\beta_l|$ changes in the optimization process in Fig. 12. Empirically, we observe that $|\beta_l|$ takes __smaller magnitudes__ during the early and intermediate iterations—precisely when escaping shallow local optima is relevant—while __larger magnitudes emerge only in the final refinement phase__, consistent with the intended design of NGA. This aligns with the theoretical role of Theorem 1, which analyzes updates around a stationary point rather than global dynamics across all layers.
>
> We have clarify this distinction in the revised manuscript (Appendix G.3) and include an additional temporal visualization of $|\beta_l|$ to demonstrate that the small-$|\beta_l|$ regime indeed appears during the iterations where the local analysis applies.
>
> > Q3
>
> We acknowledge the reviewer’s concern regarding potential instability from the product parameterization. In our experiments, we did not observe noticeable gradient instability. To ensure robustness, we applied gradient clipping, which proved sufficient to keep the optimization stable throughout training.
>
> > Q4
>
> The choice of the product form is supported by both __empirical performance and theoretical insights__.
>
> Empirically, our ablation study in Appendix F compares several alternative parameterizations. The product parameterization consistently yields better  performance, which motivated its adoption.
>
> Theoretically, Proposition 2 shows that the product structure allows $|\beta_l|$ to adaptively adjust the effective learning rate and naturally induces an accelerating effect on the magnitude of updates to $|\beta_l|$. These properties are aligned with the desired behavior of NGA.
> For these reasons, we chose the product form as it provides both practical benefits and theoretical justification.

---

### Author Response · Authors · 2025-11-30
**General Response**

Dear Reviewers and ACs,

__We would like to sincerely thank all the reviewers for their valuable feedback, and thank the ACs for the behind-the-scenes efforts.__ We are deeply grateful for your thoughtful assessment of our manuscript and for the constructive suggestions provided throughout the review process. Your expertise and insights have played an essential role in improving the quality, clarity, and presentation of our work.

__We are encouraged by the reviewers’ positive assessments of our contributions.__ We appreciate that Reviewer iuLj found the formulation of the Association Common Graph to be a key and elegant contribution, and highlighted the theoretical rigor and the adaptive optimization mechanism. We are thankful that Reviewer 6SDK recognized the clarity of our optimization analysis, the strength and breadth of our experimental evaluation, and the favorable scalability–performance trade-offs achieved by our method. We are also grateful that Reviewer xBg1 emphasized the novelty of our methodology design, the robustness of our theoretical foundation, and the extensive empirical evidence demonstrating our method’s improvements in accuracy, efficiency, and scalability.

__We are pleased that during the discussion phase, Reviewers 6SDK and xBg1 expressed clear agreement with our rebuttal clarifications and maintained their strong positive assessments of the work.__ We especially appreciate that Reviewer 6SDK further increased the confidence score after reviewing our responses. We would also like to note that, due to the unforeseen issue, we have not yet received the final response from Reviewer iuLj. Nonetheless, many of the concerns overlap with those raised by the other reviewers, and our clarifications during the rebuttal stage have already been acknowledged by them. In particular, our discussion on unlabeled graphs was positively received by Reviewer 6SDK, and our explanation regarding the computational overhead was affirmed by both Reviewers 6SDK and xBg1. For the remaining concerns raised by Reviewer iuLj, we have also provided the corresponding experimental evidence, explanations, and clarifications in the rebuttal.

__We believe all your suggestions have played an important role in improving the quality of our paper.__ Specifically, for additional experimental results and areas that all you indicated require further clarification, we have:
- provided experiments on unlabeled graphs and a variant method equipped with structural encoddings (see Response to W1&Q2 of Reviewer iuLj)
- updated the time and space complexity analysis in Appendix C.3 (see Response to W2 & Q1 of Reviewer iuLj and Q5 of Reviewer 6SDK)
- provided visualization of how the learned parameters change in the optimization process in Appendix G.3 (see Response to W3 of Reviewer iuLj)
- provided parameter analysis of learning based baselines (see the last Response to Q2 of Reviewer 6SDK)
- added a discussion on label selection guidelines in Appendix I (see Response to Q3 of Reviewer 6SDK)
- included a discussion acknowledging a line of previous works on parameterization in Appendix D.2 (see Response to Extra Q of Reviewer 6SDK)
- provided additional clarification regarding interpretability in Section 4.2.1 and Appendix B (see Response to W2 of Reviewer xBg1)
- provided comparisons with unsupervised methods for graph matching (see Response to W3 of Reviewer xBg1)

__Finally, we sincerely thank you for your time, effort, and positive feedback. We truly value all of your comments, which have been instrumental in further improving the quality of our paper.__

Best Regards,

The Authors

---

### Meta-Review · Area_Chair_gyoj · 2026-01-03

**Summary:**

The paper proposes Neural Graduated Assignment (NGA), a novel neural approach to the Maximum Common Edge Subgraph (MCES) problem. It reformulates MCES as a Quadratic Assignment Problem using an Association Common Graph (ACG) to ensure structurally valid solutions.

Reviewers were positive, praising the method's novelty, theoretical rigor, and scalability. The ACG construction was highlighted as elegant and effective, while the theoretical analysis—covering convergence, escaping local optima (Theorem 1), and the product parameterization (β_l = W_1^T W_2)—was considered a particular strength.

In addition to the reviews, the Area Chair notes related work that employs deep learning for subgraph tasks:
Xuanzhou Liu, Lin Zhang, Jiaqi Sun, Yujiu Yang, Haiqin Yang. "D2Match: Leveraging Deep Learning and Degeneracy for Subgraph Matching." ICML 2023, PMLR 202:22454-22472.

Although D2Match targets subgraph isomorphism rather than MCES, it shares the spirit of combining deep learning with structural techniques. The authors are encouraged to briefly contrast NGA with such methods to further highlight its distinctive contributions.

Overall, the innovation, theoretical depth, and empirical performance make this a strong paper suitable for acceptance. Minor revisions are needed to strengthen the presentation.

**Reviewer Concerns:**

The rebuttal thoroughly addresses nearly all reviewer concerns via new experiments, complexity analyses, visualizations, and manuscript updates. Reviewers 6SDK and xBg1 confirmed satisfaction with the responses (6SDK increased confidence). Reviewer iuLj did not post a follow-up, but their concerns largely overlap with those positively acknowledged by others.

**Reviewer Scores:**

The rebuttal's comprehensive responses and new experiments would likely have led all reviewers to strengthen their positive assessment of the work. Reviewers 6SDK and xBg1 explicitly confirmed satisfaction during the discussion phase (with 6SDK increasing their confidence score). Although Reviewer iuLj did not provide a post-rebuttal update, their concerns were thoroughly addressed through targeted experiments, analyses, and clarifications that overlap with issues positively acknowledged by the other reviewers.

---

### Decision · Program_Chairs · 2026-01-26

Accept (Poster)